# Experimental and Numerical Simulation of Extreme Operational Conditions for Horizontal Axis Wind Turbines Based on the IEC Standard

Kamran Shirzadeh [1,2], Horia Hangan [1,3], Curran Crawford [1,4]

[1] WindEEE Research Institute, University of Western Ontario, London, Ontario, N6M 0E2, Canada
[2] Mechanical and Material Engineering, Western University, London, N6A 3K7, Canada
[3] Civil and Environment Engineering, Western University, London, N6A 3K7, Canada
[4] Mechanical Engineering, Victoria University, Victoria, V8W 2Y2, Canada

*Correspondence*: Kamran Shirzadeh (kshirzad@uwo.ca)

**Abstract.** In this study, the possibility of simulating some transient and deterministic extreme operational conditions for horizontal axis wind turbines based on the IEC 61400-1 standard using 60 individually controlled fans in the Wind Engineering, Energy and Environment (WindEEE) Dome at Western University was investigated. Experiments were carried out for the Extreme Operational Gust (EOG), positive and negative Extreme Vertical Shear (EVS), and Extreme Horizontal Shear (EHS) cases, tailored for a scaled 2.2 m horizontal axis wind turbine. For this purpose, firstly a numerical model for the test chamber was developed and used to obtain the fans' configurations for simulating each extreme condition with appropriate scaling prior to the physical experiments. The results show the capability of using numerical modeling to predict the fans' setup based on which physical simulations can generate IEC extreme conditions in the range of interest.

## 1. Introduction

Wind energy is one of the primary sources of renewable energy for mitigation of the increasing global energy demand. However, one of the basic factors for this market to thrive is a continued reduction of the levelized cost of electricity (LCOE), which is enhanced by ensuring the life time of the wind energy systems is reliably long (Ueckerdt et al., 2013). Having a long life cycle for these energy systems dramatically increases the probability of them encountering various extreme weather and wind conditions. Therefore, the design of wind energy systems must consider extreme environmental conditions with statistically accurate return periods. The International Electrotechnical Commission (IEC) has some deterministic design codes for commercial Horizontal Axis Wind Turbines (HAWT) in operating conditions, specifically in the third edition of the IEC 61400 part one (IEC, 2005). These extreme models are relatively simple and are not able to capture the true coherent turbulent wind characteristics (Cheng and Bierbooms, 2001; Hansen and Larsen, 2007; Wächter et al., 2012). This is especially true in complex terrain where the gust time evolution profiles are highly asymmetric and non-Gaussian (Hu et al., 2018). It has also motivated the most recent edition of the IEC standard (IEC, 2019) to utilize statistical methods for characterizing extreme gust event performance and extrapolation of load cases. This has been enabled by computational resources to analyse wind energy systems in dynamic wind environments to expand their external condition models. However, the third edition of the IEC standards was used in the work presented here as an initial step towards gust experimentation and represents an incremental

development of a gust loading experimental capability. Progressing to a stochastic experimental approach, is left for future work and will be very challenging.

One of the extreme cases in the standard is the Extreme Operational Gust (EOG). A gust is defined as a sudden increase in velocity over its mean value, which is a transient feature of a turbulent wind field (Burton et al., 2011).These turbulent features in the Atmospheric Boundary Layer (ABL) depend on topography, surface roughness, up-stream obstacles, thermal

stability (Suomi et al., 2013) and mesoscale climactic systems such as thunderstorms and downbursts (Chowdhury et al., 2018). In theory, for different applications, there are various simplified models of gust based on a peak factor and the whole rising and falling time in the wind speed. The peak factor is the ratio of the peak velocity (maximum or minimum) and the average wind speed. Wind gusts can happen over various length and time scales in nature. The most damaging gusts for any type of structures are the ones that have the same length scale as the structure that can envelope the whole structure (Hu et al., 2018).

Smaller gusts, relative to wind turbine size can induce dynamic stall and the gust slicing effect (i.e. recurring high loads as the blade slices through the spatial/temporal gust region several times). The wind gusts also can cause intermittencies in the power output of wind turbine generators. For a small electricity network, these fluctuations in power generation can cause serious problems (e.g. unstable grid voltage and frequency) for managing power transmission and distribution (Anvari et al., 2016; Estanqueiro, 2007). The worst case in both terms of the grid stability and the loading on the turbine is when  the gust peak

speed is higher than the wind turbine cut-out speed (i.e. a specific speed that turbine comes to complete parked position for safety reasons, usually about 25 m/s), which, if prolonged enough, can cause the control system to abruptly stop the wind turbine (Hansen, 2015). From an aerodynamic point of view, gusts can result in undesired acceleration of the rotor and drivetrain. The most reasonable solution is usually an adjustable generator load or blade pitch angles after detection of the gust for modern wind turbines (Pace et al., 2015; Lackner and Van Kuik, 2010). Developing LIDAR technology can make a

substantial contribution in controlling the wind turbine by measuring the wind field upstream, thereby giving enough time for the control system to react properly (Bossanyi et al., 2014; Schlipf et al., 2013).

In addition to uniform gusts, the standard specifies deterministic Extreme Vertical and Horizontal Shears (EVS, EHS). These Extreme Wind Shears (EWS) can induce asymmetric loads on the rotor which are in turn transferred into the whole structure. The vertical shears can induce tilting or out-of-plane moments on the rotor and nacelle (Micallef and Sant, 2018). In

a positive vertical shear, the blade moving at higher height could experience stall while the one moving at lower height will experience a reduction in overall angle of attack relative to design condition (and vice versa for negative vertical shear) (Sezer-Uzol and Uzol, 2013). If the shear is extreme enough, the blades may experience a phenomenon known as dynamic stall (Hansen, 2015; Gharali and Johnson, 2015). All these phenomena together will result in high fluctuations in power generation, as well as highly dynamic fatigue loads on the structure (Jeong et al., 2014; Shen et al., 2011). The effects of horizontal shear

are similar to vertical shear in terms of power performance and blade fatigue loads. However, EHS also induces yaw moments. These transient shears can happen for similar reasons as uniform gusts, but mostly happen within wind farms, where the downstream wind turbines are partially  exposed to the wakes of other operating turbines (González-Longatt et al., 2012; Thomsen and Sørensen, 1999).

The IEC defines a classification for commercial wind turbines based on a reference wind speed and turbulence intensity,
in a way that covers most on-shore applications (IEC, 2005). The Turbulence Intensity (TI) is defined as the ratio of the
standard deviation of wind speed fluctuations to the average wind speed both calculated in 10 min intervals. TI levels of 16%,
14% and 12% correspond to the A, B and C reference turbulence classes ($I_{ref}$). For velocity references ($U_{ref}$), 3 classes have
been defined (I, II, III) with 50, 42.5, 37.5 m/s as reference wind speeds, with one further class for special conditions (e.g.
offshore and tropical storms) which should be specified by the designer. These reference velocities are used to calculate
parameters related to the turbine external conditions. For example, the standard mean value of the wind speed over a 10 min
interval based on the turbine class is $0.2 \, U_{ref}$. An extreme wind speed model as a function of height ($Z$), with respect to the
hub height ($Z_{hub}$) with recurrence periods of 50 years ($U_{e50}$) and 1 year ($U_{e1}$), is defined as follows:

$$U_{e50}(z) = 1.4 U_{ref} \left(\frac{Z}{Z_{hub}}\right)^{,0.11}$$

$$U_{e1}(z) = 0.8 U_{e50}(z),$$

(1)

This definition is used for calculating the gust magnitude of the EOG.

The design stream-wise turbulence standard deviation ($\sigma_u$) is defined by a normal turbulence model:

$$\sigma_u = I_{ref}(0.75 U_{hub} + b); \quad b = 5.6 \, \frac{m}{s},$$

(2)

$U_{hub}$ is the average wind velocity at the at hub-height and $b$ is a constant. Accordingly, the hub height gust magnitude
($U_{gust}$) is given as:

$$U_{gust} = min\left\{1.35(U_{e1} - U_{hub}); 3.3\left(\frac{\sigma_u}{1+0.1\left(\frac{D}{\Lambda_u}\right)}\right)\right\},$$

(3)

Considering t as the instantaneous time and $t = 0$ as the beginning of the gust, the uniform EOG as function of time is
defined as:

$$U(t) = \begin{cases} \overline{U_{hub}} - 0.37 U_{gust} \, sin\frac{3\pi t}{T}\left(1 - cos\frac{2\pi t}{T}\right); when \quad 0 \le t \le T, \\ \overline{U_{hub}}; \quad when \, t > T \, or \, t < 0, \end{cases}$$

(4)

T is the duration of the gust, specified as 10.5 seconds. $D$ is the diameter of the rotor, and $\Lambda_u$ is the longitudinal
turbulence scale parameter which is a function of the hub height:

$$\Lambda_u = \begin{cases} 0.7 Z_{hub} & for \, Z_{hub} \le 60 \, m, \\ 42m & for \, Z_{hub} > 60 \, m, \end{cases}$$

(5)

The EVS and EHS have similar equations that can be added to or subtracted from the main uniform or ABL inflow. The
EVS as function of height and time can be calculated using the equation (6).

$$U_{EVS}(Z,t) = \begin{cases} \left(\dfrac{Z - Z_{hub}}{D}\right)\left(2.5 + 0.2\,\beta\sigma_u\left(\dfrac{D}{\Lambda_u}\right)^{0.25}\right)\left(1 - \cos\left(\dfrac{2\pi t}{T}\right)\right); when \;\; 0 \le t \le T, \\ 0 \quad ; when\; t > T\; or\; t < 0, \end{cases} \tag{6}$$

$\beta$ is a constant with value of 6.4 and T is 12 s in the EWS. The peak factor of the EOG decreases with increasing size of the turbine or decreasing hub height, and vice versa for the EWS based on these equations.

Along with more common steady state experiments (Snel et al., 2007; Sørensen et al., 2002), developing  transitory flow field experiments have attracted the interests of researchers during the past few decades (Lancelot et al., 2017; Ricci et al., 2017) to evaluate the various computational techniques or to directly investigate complex phenomena in different applications. In the wind energy field, some efforts have been made to produce gusts; for example, using active grids (Petrović et al., 2019; Wester et al., 2018) and a chopper mechanism (Neunaber and Braud, 2020). Developing these unsteady flow fields basically comes down to the experiment targets and the available wind tunnel facilities. In this study, the generation of the EOG and the EWSs unsteady flow fields with relevant scaling (customised for a 2.2 m scaled HAWT) using 60 individually controlled jet fans in the WindEEE dome are considered. This work presents a new numerical model of the WindEEE dome test chamber which can be used to predict fan settings for any custom steady or unsteady 2D flow fields before the physical experiment, and the capability of this facility to physically generate the gusts and shears similar to IEC standard during experiments. The focus of this paper is just on the time evolution of the simulated extreme conditions' flow fields which is a prologue for future experiments including an actual HAWT model.

The paper is organized in three sections beside the introduction and it is as follows. Section 2 details the development of the numerical model for the WindEEE test chamber which was used to obtain the fan setups to use in physical simulation of the gusts. This section also provides a length and time scaling of the gust which based on that the target gusts for experimental campaign are introduced. Section 3 presents the results from velocity measurements at the test section in two parts, firstly the steady shears to assess the accuracy of the developed numerical model to simulate the shear layers and secondly the final transient simulated gusts and their comparison with IEC standard. Section 4 provides some conclusions.

## 2. Methodology

### 2.1. WindEEE Dome

The physical experiments were conducted in the WindEEE Dome at Western University, Canada. This is a versatile facility that can be run at  different modes for creating various  non-stationary wind systems (Hangan et al., 2017). It has an inner test chamber with a 25 m diameter hexagonal footprint and 3.8 m height. It has a total 106 fans, including 60 fans installed on one wall and 40 fans over the other five peripheral walls. There are also 6 larger fans in a plenum above the test chamber which are mostly used for generating 3D flows like tornados and downbursts. The test chamber is in turn surrounded by an

outer shell. The dome inner shell/test chamber along with outline of the outer shell with the flow path in the closed-circuit 2D flow mode (e.g. ABL, shear flows and etc) are presented in Figure 1a. In 2D flow mode, the louvers at the top and peripheral sides of the test chamber are closed and the flow is energized only by the 60 fans, then it reaches to the test section (center of the test chamber) and then exits the test chamber through the mesh of the wall at the opposite end, then recirculating over the top while passing through the heat exchangers, and finally back to the 60 fans' inlet. Each fan is 0.8 m in diameter with 30 kW nominal maximum power. In order to reach higher velocities and better flow uniformity characteristics at the center of the test chamber, a two-dimensional contraction can be setup to streamline the flow as shown in Figure 1b.

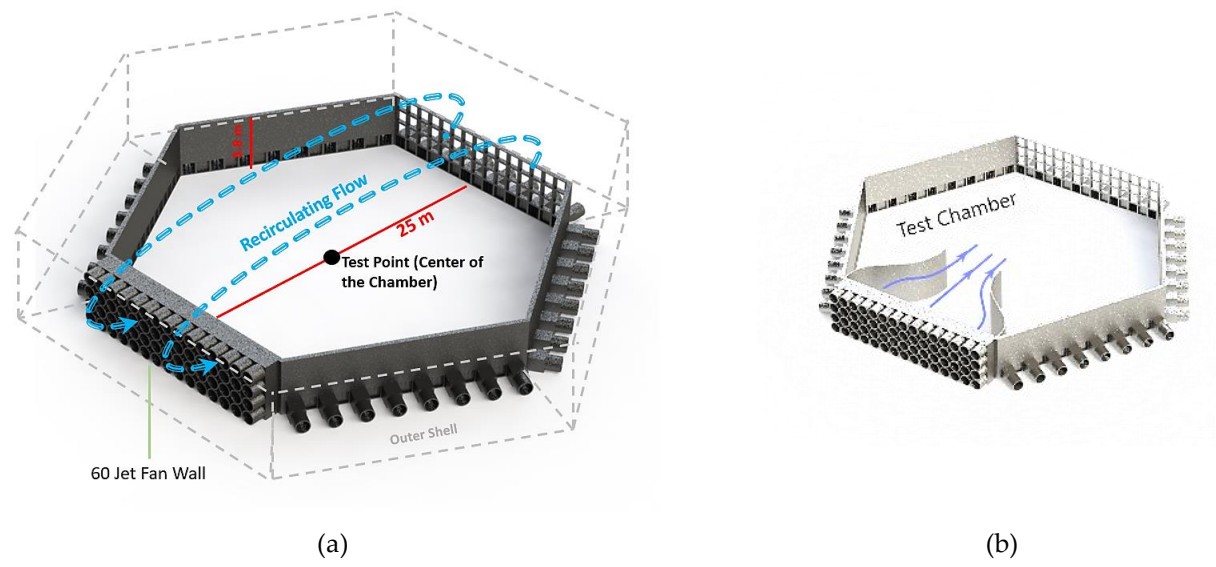

(a)                                                                                          (b)

**Figure 1. A brief geometry of the WindEEE dome, (a) the test chamber with outline of the outer shell along with the flow path in closed-circuit 2D flow mode, (b) the test chamber with contraction walls**

The power set-points of the 60 fans can be adjusted by the software as fast as 2Hz. However, this does not imply the fans themselves can throttle from 0% to 100% power at 2 Hz (due to rotational inertia of the fans' rotors and electrical current filtering it takes ∼3 s for the fans to adjust).

Another feature is that the fans are equipped with adjustable inlet guide vanes which can regulate the amount of flow rate through the fans. These vanes can be adjusted uniformly from 0% open (close) to 100% open (Figure 2). They can also be adjusted dynamically by setting an actuation frequency, duty cycle and an initial position. The actuation frequency specifies the time between two cycles, while the duty cycle specifies the duration of an individual cycle specified as a percentage of the time between two successive cycles. All these features allow the generation customizable dynamic flows.

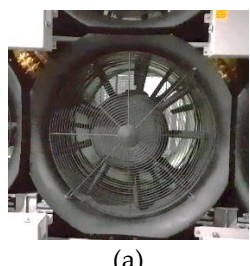 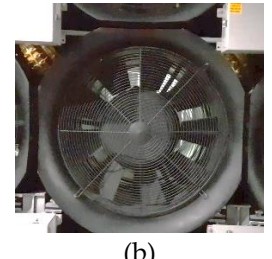

(a)               (b)

**Figure 2. The adjustable vanes at the inlets of the 60 fans wall, (a) 100% open vanes, (b) 70% open vanes**

135         **2.2. Numerical Flow Analysis Setup and Tuning/Validation**

In order to have a better understanding of the flow field in the test chamber, a numerical model for the test chamber was created using the commercial Star-CCM+ CFD software, which helped to predict the fan power setups for different scenarios prior running the experiments.

For this purpose, four simplified symmetrical domains of the test chamber were generated to save considerable CPU time

140   as listed at Table 1. As this table outlines, the domains V and V-c were used for simulating EOG, EVS and ABL flows; domains H and H-c were used for simulating EHS.

**Table 1: The symmetrical domains of the test chamber used for simulating different cases**

| Picture of the Domain | Application | Domain ID |
|---|---|---|
| | Simulating ABLs and EVS & EOG and tuning the boundary conditions parameters | V |
| | Simulating ABLs and EVS & EOG with contraction walls and tuning the boundary conditions parameters | V-c |
| | Simulating EHS | H |

| | | |
|---|---|---|
| 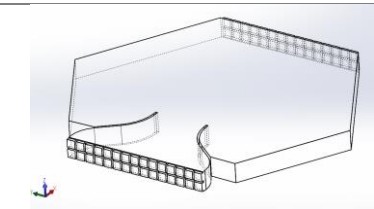 | Simulating EHS with contraction walls | H-c |

In order to discretize the domains, three mesh setups (M1, M2 and M3) were considered for the polyhedral automated mesh function, built-in Star-CCM+ software. The general details for the generated grids are presented in Table 2. For all the cases, 5 prism layers with a total thickness of 0.05 m and with stretching of 30 % at the solid walls with minimum of 4 elements in the gaps were used; the surface curvature and surface growth rate were left at their default values (6 degree and 20% respectively) with no specified mesh density in the domains. In addition, in domains with contraction walls a custom control refinement on the surfaces of the contraction walls was used to create elements half of the general base size. The fans were modelled as squares with individual velocity inlet boundary conditions. The outflow grid on the opposite wall was treated as uniform pressure outlet. All other surfaces were treated as no-slip walls. Due to broad range of the Reynolds number across the domain, controlling the wall y+ was challenging. Therefore, for modelling the Reynolds stresses in the RANS equations, two-layer K-epsilon (k-ε) turbulence model was chosen.

The next step was to calibrate the boundary condition parameters based on the previous experiment data that were available for scaled ESDU ABL profiles both with and without contraction walls (Hangan et al., 2016). The simulated fan powers were then adjusted to reach the desired average velocity profiles at the test section to match the existing experimental data. The M1 setup at domain V and V-c were used for preliminary tuning of the input values at the inlets and the outlet boundary condition parameters in order to get the best match with the available data at the test section. The best results corresponded to an inlet turbulence intensity of 8% with length scale of 0.2 m and the outlet boundary set as a pressure outlet with uniform zero-gauge pressure, 1% turbulence intensity and 0.05m length scale. Working at full power, the fans can generate 13 and 31 m/s of uniform wind velocity at the test section without and with contracting walls respectively. At the end the simulation results showed that the full fan powers corresponded to a 16.5 m/s inlet boundary velocity. The fan power set-points were then simplified as a linear interpolation between 0 and 16.5 m/s for the velocity inlets.

**Table 2: Detail of grid sizes for each domain**

| Grid name tag | M1 | M2 | M3 |
|---|---|---|---|
| Number of Cells for Domain V (Million) | 1.41 | 2.53 | 5.52 |
| Number of Cells for Domain V-c (Million) | 2.37 | 3.72 | 6.75 |
| Number of Cells for Domain H (Million) | N/A | 1.93 | N/A |

| | | | |
|---|---|---|---|
| Number of Cells for Domain H-c (Million) | N/A | 3.00 | N/A |
| Base size (m) | 0.1 | 0.08 | 0.06 |

The mesh independency check was defined by the incrementally refined grids M1 to M3 using the velocity profiles at the test section for the ABL profiles which have different fan power set points for each row (Figure 3). For low speed setup (without contraction) they were at 50, 70, 70 and 50% from bottom row to top (Figure 3a); in the setup with contractions, the fans are at 50, 65, 75 and 75% (Figure 3b). The velocity profiles from the CFD results were defined by a vertical probe line passing through the center of the test chamber with 40 points over the entire height of the chamber.

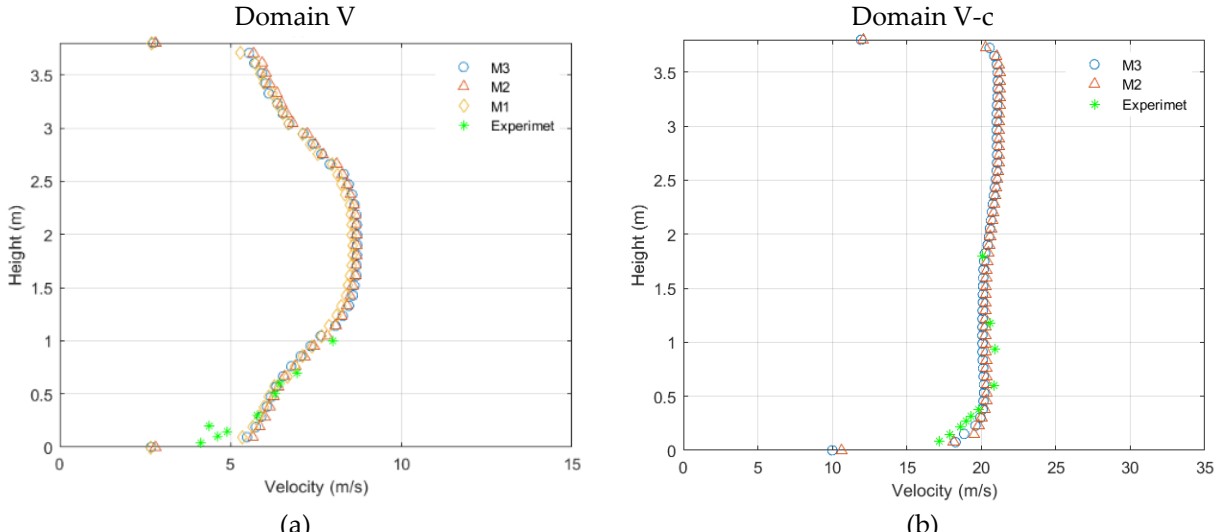

(a)

(b)

**Figure 3. The mean ABL velocity profiles at the test section for different mesh setups comparing with the experimental data** (Hangan et al., 2016)**, (a) low speed (without contraction) and (b) high speed (with contraction) mean velocity vertical profiles**

Figure 4a &b show the relative errors between velocities at each height; the largest disconformities between different mesh setups occur close to the wall which for this research is not the most important region. The more critical region for the present experiments is at the middle heights where the wind turbine rotor will be located. That being said, even the M1 setup has an acceptable range of error (∼1%) at mid-height. However, the M2 mesh setup was chosen as the best compromise of computation speed and accuracy.

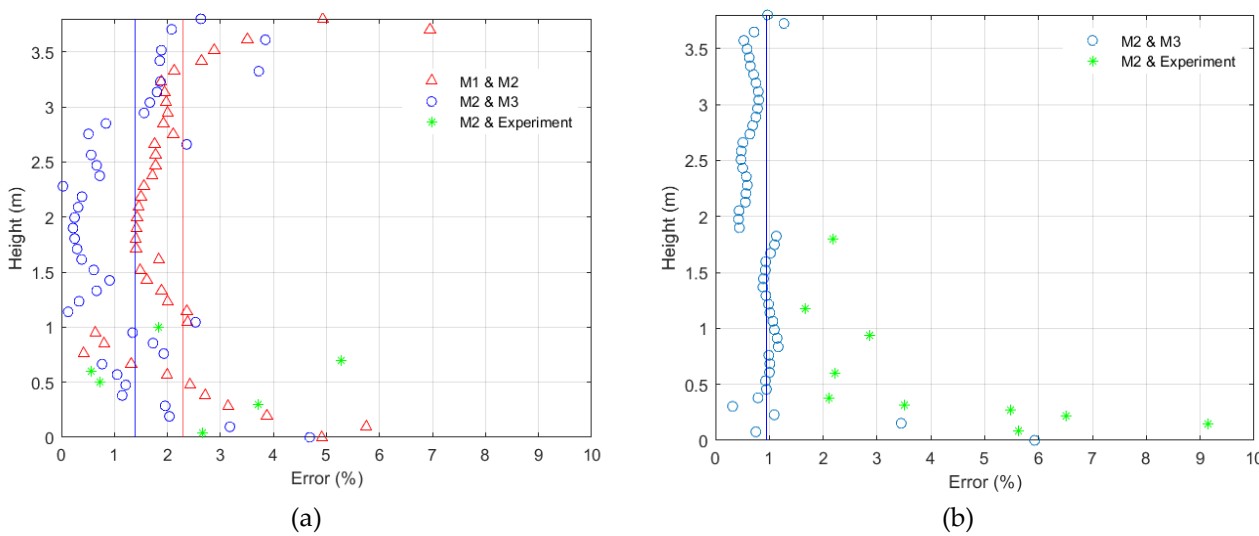

Domain V                                                        Domain V-c

(a)                                                                  (b)

**Figure 4. The relative errors for (a) low speed velocities and (b) high speed velocities, the solid lines are the mean value for the errors over the whole height**

The discrepancy between the CFD simulation (M2) and the experimental data also increases close to the wall. This error is rooted in uncertainty of the implemented turbulence model and relative course mesh size close to the wall in the numerical model. Nevertheless, they are in an acceptable range of engineering applications (under 10% of relative error). A picture of discretized domain V-c with the M2 grid is shown in Figure 5.

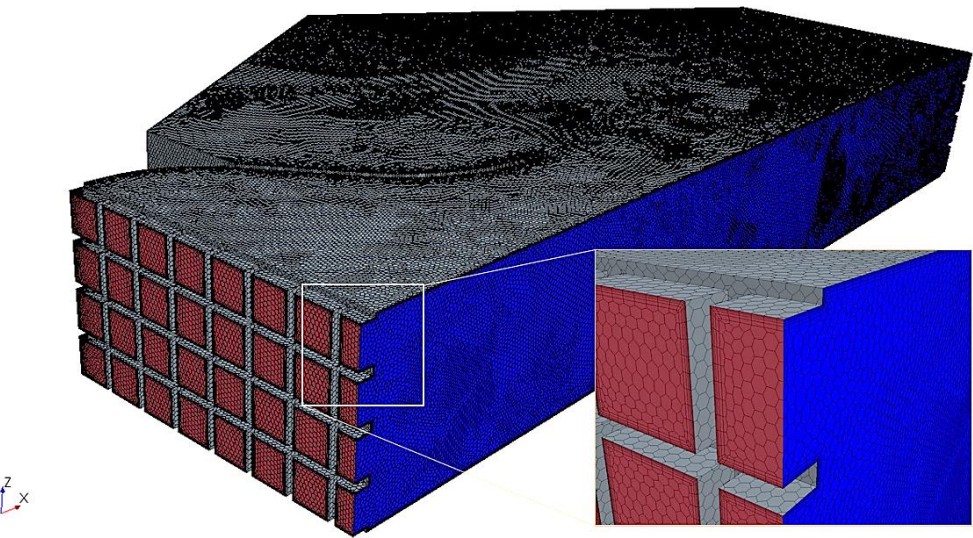

**Figure 5. The M2 grid for the V-c domain**

As described, this numerical model has been developed based on a set of steady ABL experimental data. The first application of it was to generate a calibration table that related the steady fan power set-points to the mean velocity magnitudes and profiles at the test section. This table was used to predict the fans' powers in generating the EOG. For simulating the EWSs, only the peak stages of these extreme events were considered for modelling, again in steady condition in order to obtain

the fan setups at the peak of the corresponding wind shear event (see 3.1). These numerical simulations neglect the closed loop flow recirculation dynamics in the dome. Nevertheless, it produces a reasonable prediction of the fan setups for a specific flow field in a reasonable amount of time.

### 2.3. Experimental setup for velocity measurements

The velocity measurements were obtained with seven cobra probes. These robust probes (TFI Ltd., 2011) are capable of

195 measuring the incoming airflow velocity within a cone shape of 45° with up to 10 kHz sampling frequency. Each probe has 4 pressure tabs at the head (0.5mm each) and is able to measure three velocity components with measuring range from 2 to 45 m/s with $\pm0.5$ m/s and $\pm1°$ pitch and yaw accuracy up to approximately 30 % turbulence intensity. In this study, the average stream wise wind velocity was 5 m/s; therefore, all the presented wind measurements have $\sim$ 10% accuracy in average.

Two different setups for velocity measurements were used; vertical and horizontal arrangements (Figure 6a & b). The

200 sampling duration was 60 s with sampling frequency of 2000 Hz for each measurement run. The sampling duration was considered long enough compared to the 5 s extreme events to check for any unexpected perturbation in the flow field due to running the experiments in closed loop mode as the flow recirculates, by considering the flow recirculation path ($\sim25 + 4 + 25 + 4 = 58\ m$) and the average wind speed ($\sim5\ m/s$) which give recirculation time of 12 second. The precise turbulence

characteristics is not the major objective of the current study so the sampling frequency was chosen based on previous studies
in this facility (Refan and Hangan, 2018; Romanic et al., 2019). In each extreme event multiple actuation times for modulating
either the fan powers or the IGVs were considered; this study presents the best results compared to the target extreme events
from an individual test run. More detail about cobra probes connection is presented in Figure 6d (just one of the probes is
presented in this figure). The blue and yellow arrows are used for annotation of connections and equipment respectively. For
correct measurements these probes need a static reference pressure. Therefore, all of them were connected to the static pressure
side of a pitot tube via a manifold. The pitot tube was installed close to the cobra probe E in the middle of the array. Each
cobra probe interface box has four cobra probe and eight analog input channels (the analog channels were not used). Therefore,
two interface boxes were used which were connected with a specific synchronizing cable. Then each of these boxes were
connected to the same A/D card via USB cables. The A/D card then was connected to a laptop that had the required TFI
software installed.

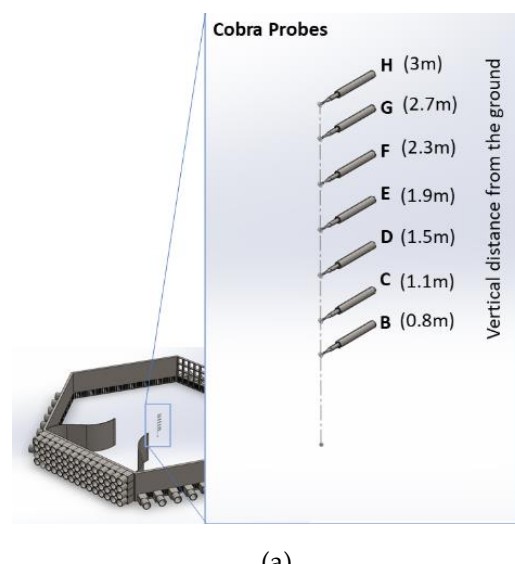

(a)

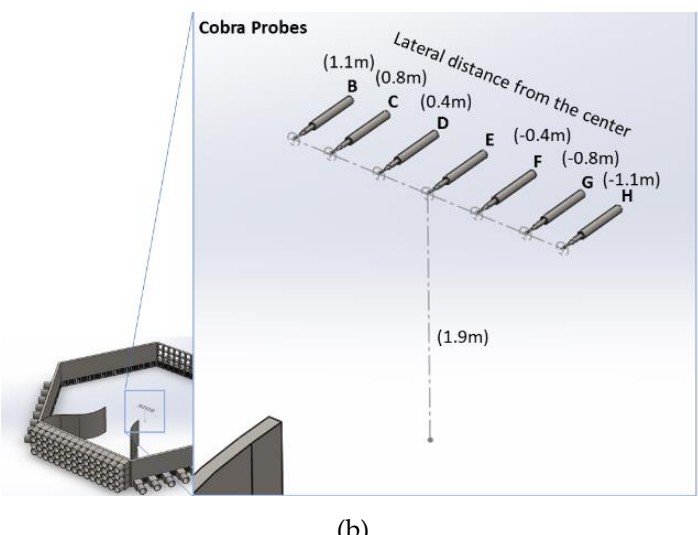

(b)

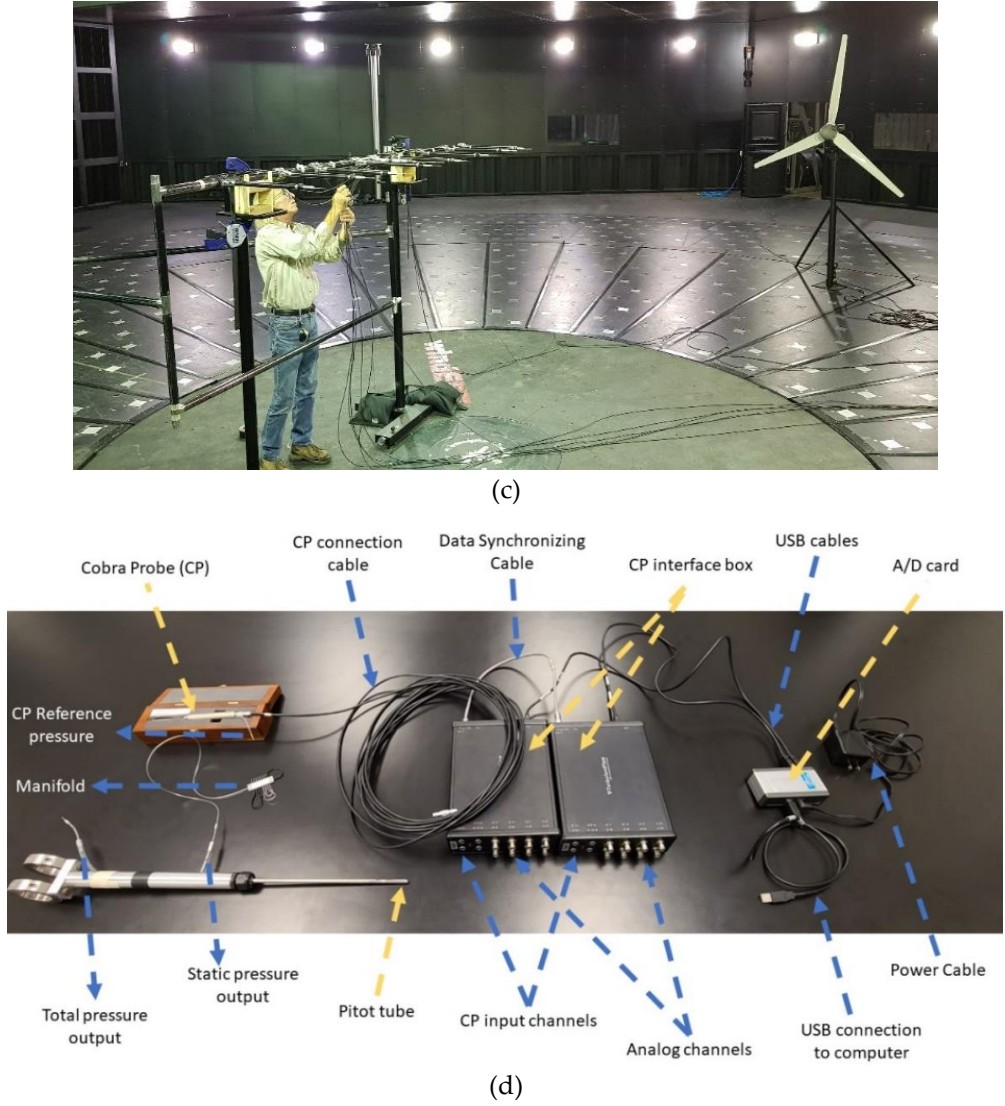

(c)

(d)

**Figure 6. The arrangement of cobra probes based on the dimension of a 2.2m diameter HAWT for (a) vertical and (b) horizontal measurements at the center of the test section, (c) Setting up the 7 cobra probes in a horizontal arrangement at the test section, (d) Cobra probe connection details**

The locations of the probes were chosen based on the dimension of the available wind turbine in the facility. This turbine has a 2.2 m diameter with adjustable hub height, chosen as 1.9 m (Refan and Hangan, 2012). This entire paper is dedicated just to the development of the flow field. Investigation the effect of these unsteady wind conditions on the turbine is left for future work.

## 2.4. Gust length and time scaling

The time durations of the extreme events (T), as mentioned earlier, are 10.5 s for EOG and 12 s for EWS (IEC, 2005). Subsequently, the gust durations correspond to 3 to 4 complete rotor revolutions periods for full-scale turbines (which usually have angular speed of 15-18 RPM in 10 m/s average wind speed). Usually, for a scaled wind turbine in the wind tunnel 4 rotor revolutions happen on the order of a second at the nominal operating condition. This gust time scale would be impossible to

simulate at WindEEE facility given the physical limitations of the hardware. Therefore, by assuming that the time scale of the gust is equal to propagation time of 4 loops of a blade tip vortex downstream in the wake, the relevant gust time becomes function of turbine operating parameters and wind speed which then can be adjusted. We can calculate the propagation length and time of these vortex loops based on the definition of the Tip Speed Ratio (TSR: $\frac{blade\ tip\ speed}{free\ stream\ speed}$); assuming a uniform wake we have:

$$\Omega = \frac{\lambda U_{hub}}{R}\ [rad/s],$$

$$\Omega' = \frac{\lambda U_{hub}}{R} \times \frac{1}{2\pi}\ [rev/s], \tag{7}$$

$$U_{hub} \times \frac{1}{\Omega'} = \frac{2\pi R}{\lambda}\left[\frac{m}{rev}\right],$$

where $\Omega$ is the angular velocity in radiant per second and $\Omega'$ is in revolution per second, $\lambda$ is TSR and $R$ is the radius of the rotor; with some rearrangement the last part in equation (7) can be rewritten as follow:

$$\frac{L'}{D} = \frac{T'U_{hub}}{D} = \frac{\pi}{\lambda}, \tag{8}$$

the $L'$ and $T'$ are the length and time duration for propagation of one vortex loop in the wake. Based on the equation (8)

and our assumption, an appropriate gust time and length can be calculated from:

$$\frac{L_s}{D} = \frac{T_s U_{hub}}{D} = 4\frac{\pi}{\lambda}. \tag{9}$$

Accordingly, the scaled time duration ($T_s$) is function of TSR, free stream velocity and the diameter of the rotor. The scaled length ($L_s$) is function of TSR and diameter of the rotor (Figure 7).

If the scaled turbine works at the same TSR and free stream velocity as the full-scale commercial HAWT, the time and length scale would be equal to their geometrical scale (i.e. the ratio of diameters).

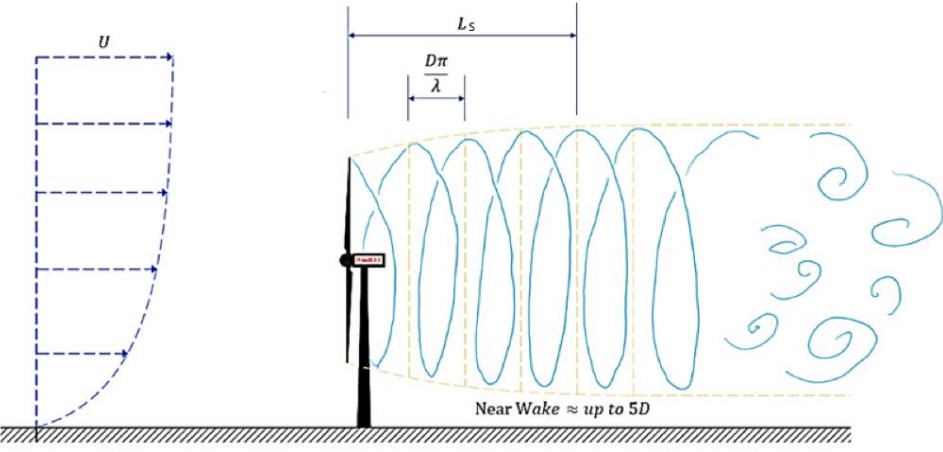

**Figure 7. Visual representation of the length and the time scale relevant to the extreme conditions with assuming a symmetric wake**

The flow behaviour in the near wake region is directly correlated to the overall performance of a HAWT. Matching the time duration of the extreme condition to the propagation of 4 vortex loops in the wake should be a reasonable comparison to the full scale in terms of variation of power and loads on the wind turbine.

       For a commercial B-III class HAWT with 92 m diameter rotor and 80 m hub height, at 10 m/s average velocity, the prescribed EOG and EVS are presented in Figure 8a & b. The time windows in these figures start and end with the extreme
events.

       The physical experiments showed that the fastest possible gust events with the required peak factor were around 5 seconds due to the hardware limitation. Therefore, to match the extreme event period to the suggested scaling assumptions, the 2.2 m scaled wind turbine should work in 5 m/s free stream velocity with operating TSR of 1.1, then it would take 5 seconds for the four complete loops of the tip vortexes generated by a specific blade to propagate in the wake. Accordingly, in all of
the simulations and experiments the hub height velocity was kept at 5 m/s. Assuming a similar B-III class for the scaled HAWT with the hub height of 2 m, the scaled extreme condition profiles are shown in Figure 8c & d; in the scaled EOG velocity should uniformly rise from 5 to ~7.8 and then back to 5 m/s in 5 seconds with ~1 m/s drops before and after the main peak relative to the average free stream velocity (Figure 8c). However, in the experiments the gusts have been simplified by not including the velocity drops (the red dashed-line in Figure 8c). This simplification stretches the actual rising and falling time
from ~ 2.5 to 5 s. Yet, this is the compromise that was made due to the hardware limitations. Hence, in this study, the target EOG has the same falling and rising time period as the scaled EWSs. The pre-post dips in the standard EOG reflect field data wherein gusts are preceded by lulls; however for the purpose of investigating peak loading during gust events, for a machine nominally operating at the mean wind speed and assumed not responding much during the lull period, it is the velocity excursion above average wind speed that is important to capture. Future apparatus design and fan control may enable execution
of pre-post lulls in prospective experiments.

In the scaled EVS the uniform velocity field transitions to a highly sheared flow (~7 m/s velocity shear over 2.2 m distance) and then back to a uniform field, again in 5 seconds (Figure 8d).

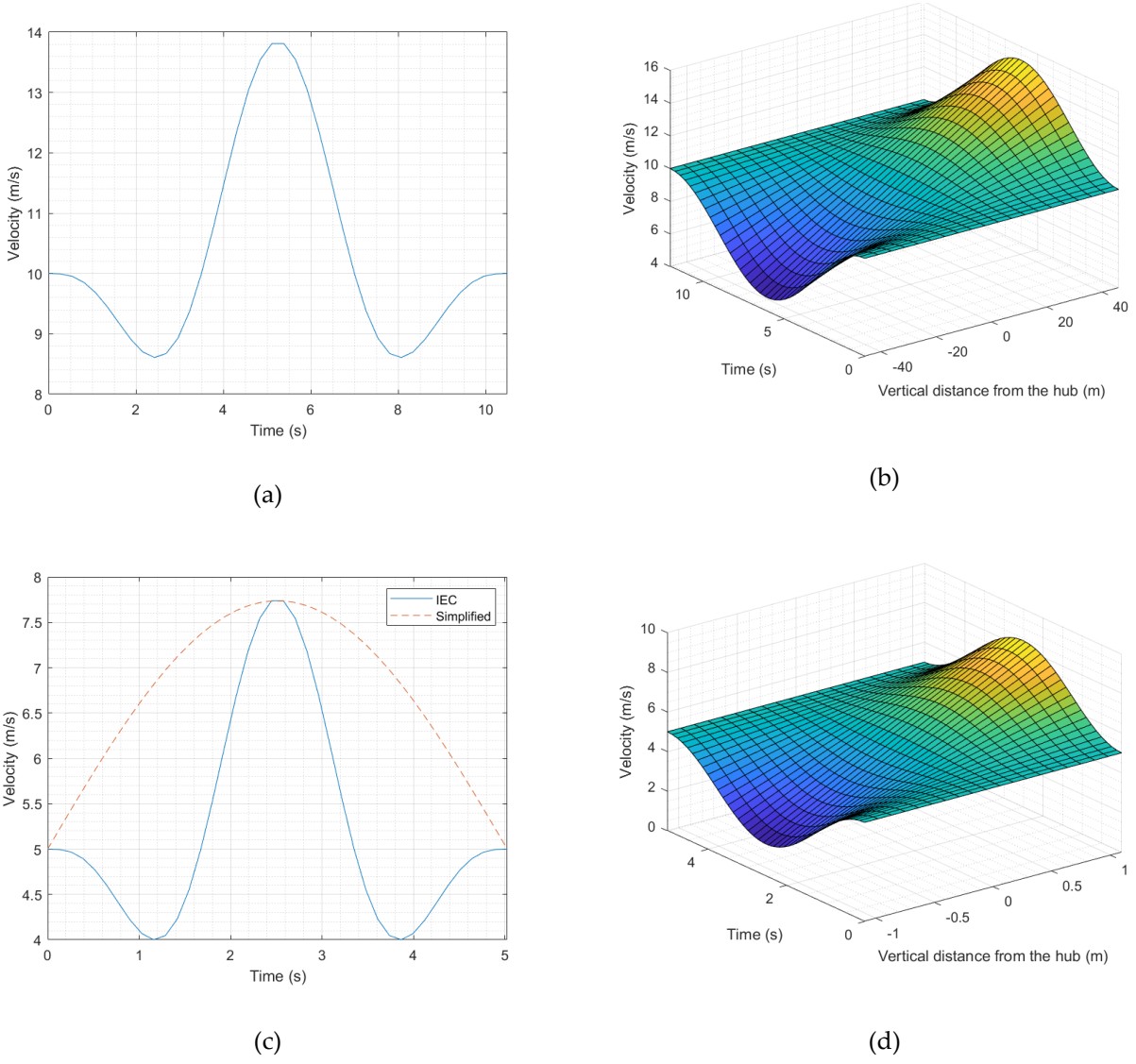

**Figure 8: The extreme operational conditions for a full scale HAWT class B-III with 92 m diameter and hub height of 80m at 10 m/s uniform wind speed compared with the scaled conditions for a B-III turbine with 2.2 m diameter and 2 m hub height at 5 m/s average wind speed (a) full scale extreme operational gust, (b) full scale extreme vertical shear, (c) scaled extreme operating gust, the solid blue line is for IEC and the simplified gust that actually was targeted is in red dashed-line, (d) scaled extreme vertical shear**

In these settings, the length and time scale ratios are 5.2 and 2.4 respectively. The Reynolds number based on the relative velocity and chord size at the 70% blade span for full scale turbine at the nominal wind speed and TSR (10 m/s and 8 respectively) is $\sim 7.5 \times 10^6$ and for the scaled turbine at our lab condition is $\sim 32.5 \times 10^3$ which gives the ratio of $\sim 230$.

## 3. Results and discussion

### 3.1. Steady wind shear

In this section, the simulation cases are all steady and just carried out for the peak stages which is the instantaneous point in time where that maximum shear occurs, as a preliminary investigation to unsteady experiment runs that are examined in the next sub-section. Using the tuned numerical model, the V-c and H-c domains were used to simulate the desired vertical and horizontal sheared flows by modulating the input velocity for the different rows and columns of the fans. The target was to match the velocity profile as similar as possible to the IEC standard for the scaled HAWT, corresponding to $\sim 7$ m/s shear while keeping the velocity at the rotor centerline 5 m/s. Figure 9 shows the fan setups using CFD for creating the desired shears which could be achieved by using only the 5 fan columns at the middle. For creating negative vertical shear, the setup presented in Figure 9a was inverted.

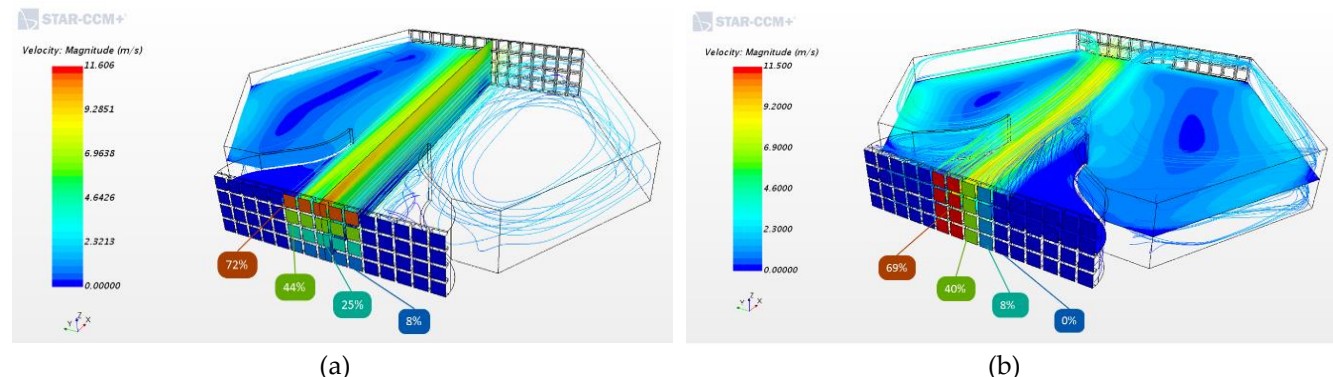

(a)                                                                                          (b)

**Figure 9. Fan setups for peak stages of extreme (a) vertical and (b) horizontal shears, prescribed for the scaled HAWT identical to full scale condition, the power set-points for each row and column are included (just the 5 columns at the middle are working)**

Using the fan setups shown in Figure 9 the physical experiments were carried out and velocity measurements made using the cobra probes. Figure 10a, b & c show the average velocity at each probe including the range of velocity fluctuations (standard deviation) compared with average velocity profile from the CFD (dashed-line) and the prescribed shear by IEC standard (yellow solid-line) for the EVS, EHS and negative EVS respectively. The high velocity fluctuations relative to the mean velocity in experiments are due to the strong vortexes that form in these highly sheared flows which increase the momentum mixing at different heights. The amount of shear that was prescribed ($\sim 7$ m/s velocity difference) is being successfully created in the tunnel for the positive vertical shear case (Figure 10a). However, for the horizontal and negative

vertical cases (Figure 10b & c) there are larger shears than desired, resulting in a ~10 m/s velocity difference. From Figure 10a & c it is clear that the lower fans work more efficiently than the upper fans (i.e. with the same value of the power set-points, the lower fans generate higher velocities). The largest disconformity exists in the horizontal shear case (Figure 10b).

The relative discrepancy between the mean velocity fields of these three experimental steady shears and the IEC are presented in Figure 10d. Accordingly, the average amount of disconformities over all of the probes are 41, 27 and 9 % for the
290 horizontal, the negative vertical and the vertical steady shears respectively. Basically, this comparison revealed the capability of the developed numerical model to predict the fan setups for simulating the EWS. As was explained in section 2.2, the numerical model is tuned just based on previously tested ABL flows while assuming similar efficiencies for all the fans, neglecting the flow recirculation in the outer shell and simplifying WindEEE test chamber geometry. The fan power values in all the test cases (steady and unsteady) are directly taken from the steady numerical model prediction results. In future, further
field adjustments are required to generate a flow field as similar as possible to the IEC prescription.

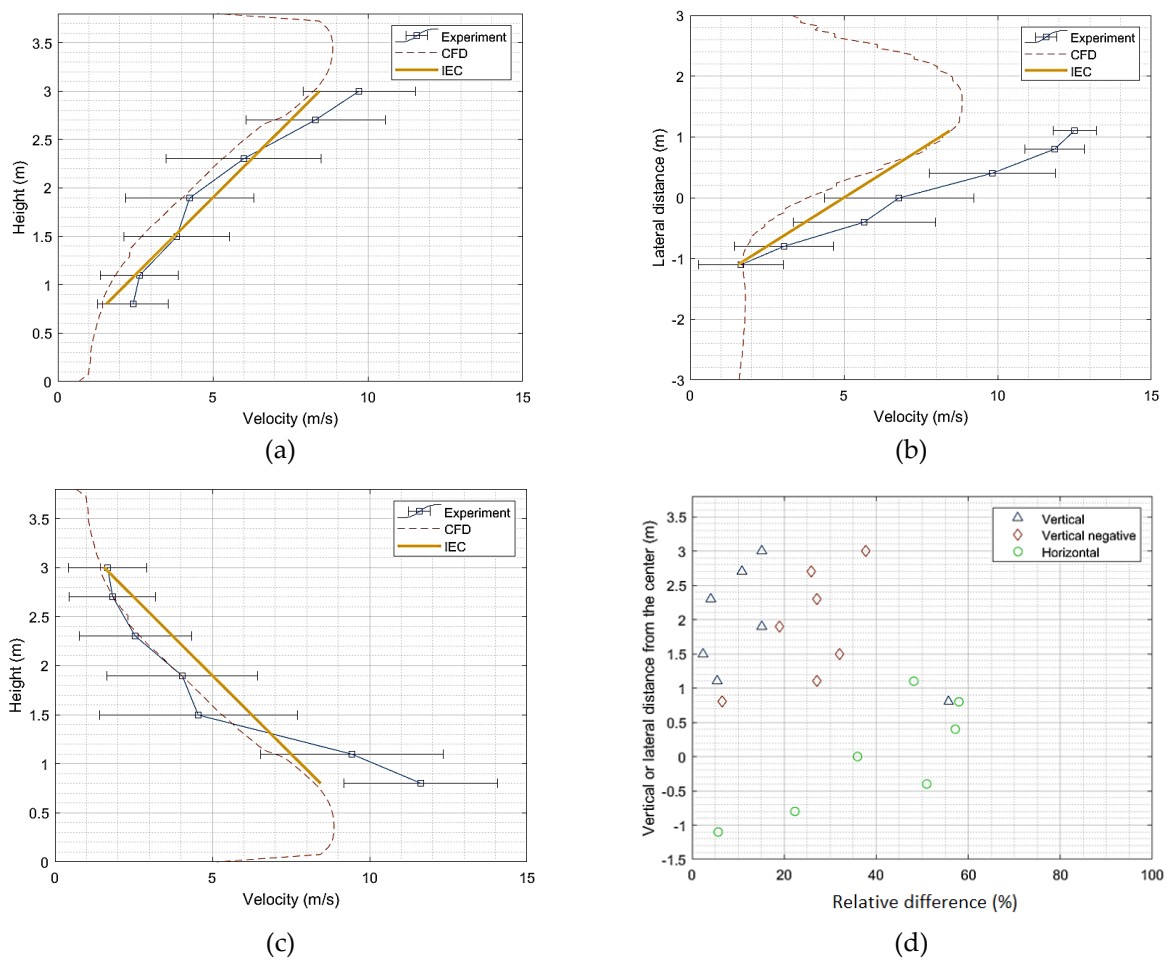

**Figure 10. CFD predictions vs experiment data for steady (a) vertical shear, (b) horizontal shear and (c) negative vertical shear, (d) the relative disconformity between the three steady shear experiments and IEC standard**

### 3.2. Unsteady experiments

For the shear cases just the five columns of the fans in the middle were working (only 20 out of 60 fans were operated). The uniform flow field before and after the shear events, generated by setting these 20 fans to 39% power. The best results in terms of the event duration, were captured when the extreme condition setups were set for 1.6 s in the actuator software (i.e. the fan powers uniformly stayed at 39% and then switched to the setup in Figure 9 for just 1.6 seconds then back to the 39% uniform). The uniform gusts were generated in two ways. The first was again by changing the power set-points of all the 60 fans together. According to the results from the CFD simulations (Domain V-c), in order to achieve the prescribed EOG, the fan power set-points should switch from 17% to 30% and back to the 17% power in the software. For the uniform gust, the best result again was obtained with setting fans to 30% for 1.6 s which resulted in ~5 $s$ uniform gust with desired peak factor. The second way of generating a uniform gust was using the IGVs while keeping fan power set-points constant at 30%. In this run, the actuation frequency of the IGVs was set at 0.05 Hz with a duty cycle of 8%, initial position of 10% open with cycling to 100% open (see section 2.1 for IGV setting definitions). In addition, in each uniform gust case, to obtain a better understanding of the uniformity of the flow field, two measurement runs were conducted using both vertical and horizontal layouts of the cobra probes (layouts in Figure 6). For processing the data all of the velocity time histories were filtered using a moving average with an averaging window of 0.2 s based on the criteria described at (Chowdhury et al., 2018).

3D pictures of the filtered turbulent wind fields for the EVS, EHS, negative EVS, EOG cases generated with changing fan powers (vertical & horizontal measurements), and the EOG generated using the IGVs (vertical & horizontal measurements) are presented in Figure 11a, b, c, d & e, f & g respectively. The average magnitude of fluctuations around the mean velocity values due to the filtration, are $\pm 0.16$ m/s for the EWS cases, $\pm 0.11$ m/s for the EOG using the fan powers and $\pm 0.41$ m/s for the EOG using the IGVs. In Figure 11a & c when the 20 fans at middle are operating, it is again evident that the fans at the top row do not work as efficient as the other fans; they could have less stable air supply than the lower rows which should be due to the tight direction change of the recirculating flow from the top. Figure 11d & f show velocity fields when all 60 the fans are operating with the contraction walls to help unifying the flow field. Figure 11b, e & g show that all of the flow fields are horizontally uniform. The data from the EOG generation with IGVs (which work in a cyclic manner) in Figure 11f & g shows the background velocity fluctuations are high relative to the EOG generation by manipulating the fans' powers in Figure 11d & e.

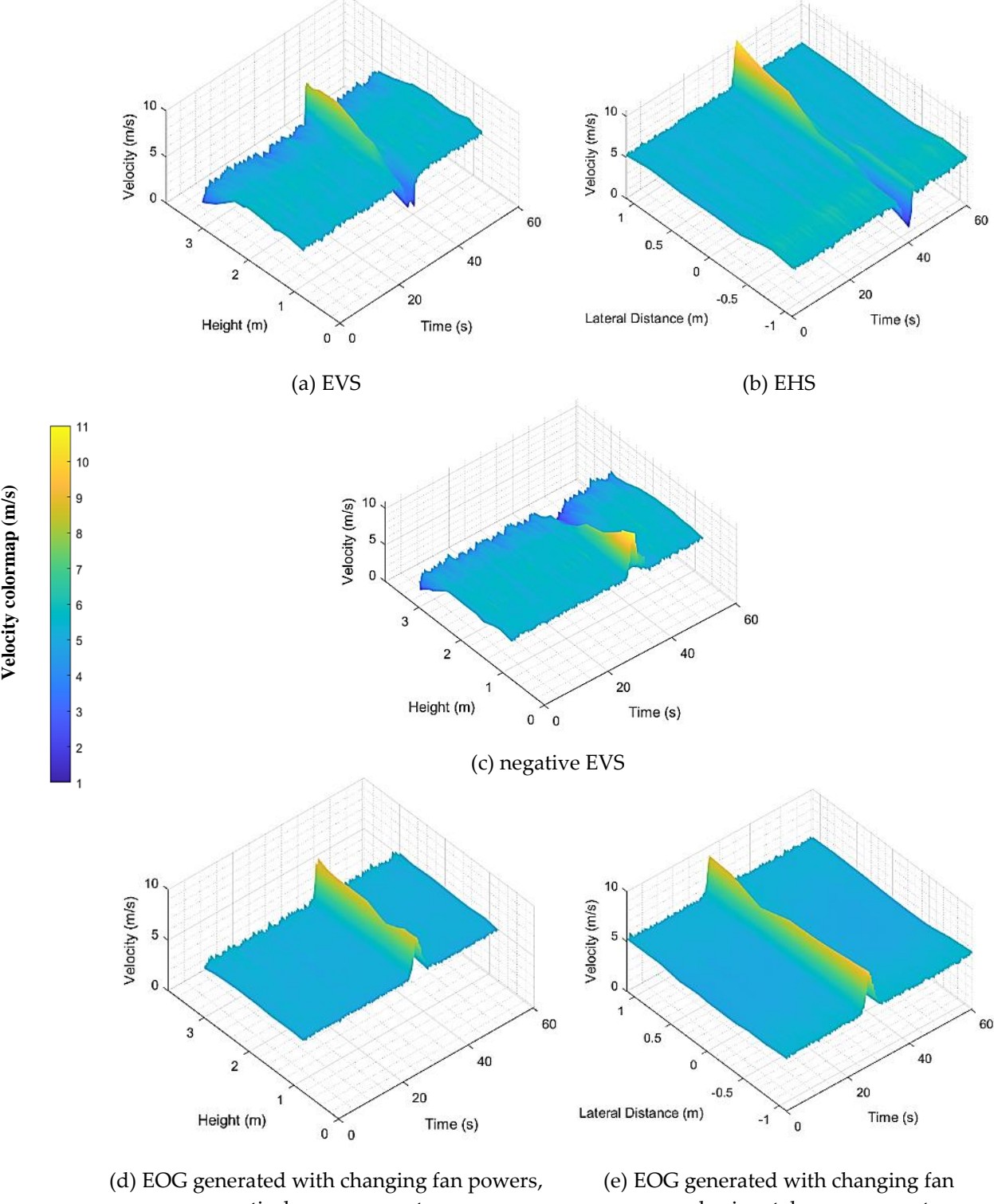

(a) EVS

(b) EHS

(c) negative EVS

(d) EOG generated with changing fan powers,
vertical measurements

(e) EOG generated with changing fan
powers, horizontal measurements

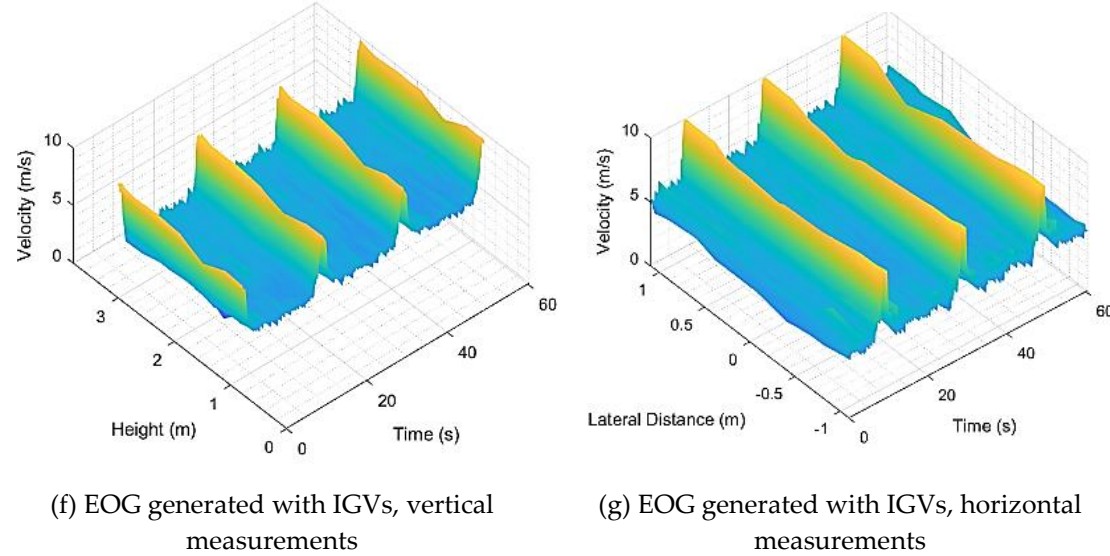

(f) EOG generated with IGVs, vertical
measurements

(g) EOG generated with IGVs, horizontal
measurements

**Figure 11. 3D pictures of the complete time history of the phased averaged (with 0.2 s averaging window) turbulent velocity field**

In order to have a better comparison of these unsteady cases with the IEC, the velocity time history extracted from the
325 cobra probes B to H (with the layout showed in Figure 6) in blue solid-lines along with the standard specifications in orange solid-lines are plotted in Figure 12 at the left columns (cases are in the same order as Figure 11). The right columns contain the relative instantaneous discrepancy of the velocity relative to the IEC prescribed velocity, normalized by the average velocity ($\sim 5 \ m/s$).

Based on data for the shear cases, at the peak stages the amount of desired shear is successfully being generated. However,
due to the difference in velocities, there is a time lag between the peaks' locations at the top to bottom heights of the EVS cases, and left to right in the EHS cases (Figure 12a, b and c).

As previously discussed, when just the 20 fans in the middle are working the lower efficiency of the top row of the fans is more noticeable at probe H  in Figure 12a & c; the velocity time history at this height and condition has more fluctuations compared to the other probes. Probe H in Figure 12d & f shows in better detail that using all the 60 fans and the contraction
walls helps homogenizing the flow field close to the ceiling (i.e. similar velocity magnitudes and fluctuations in all the time histories across the probes ). Yet, in the gust peak when the flow is highly dynamic, the insufficiency of the air supply for the top row is noticeable as the probe H in  Figure 12d demonstrates (the sudden velocity drop while the velocity in other probes are consistently increasing). Similar velocity instabilities have been observed at the same height in other experiment runs when rapid fan power changes were applied. Figure 12d & e in detail present the flow field of the EOG generated by changing the
whole 60 fans' powers. As the discrepancy time histories suggests, the generated EOGs with this method are consistent with the target simplified gust. However, the profiles are slightly asymmetric; the left sides of the gust profiles have positive curvature and the right sides have linear behaviours. This could be due to the fact that the fans do not decelerate as fast as they

can accelerate (the gust falling time is not as fast as its rising time). Active fan braking might be explored in future work to accelerate the falling gusts, instead of relying on inertia/friction.

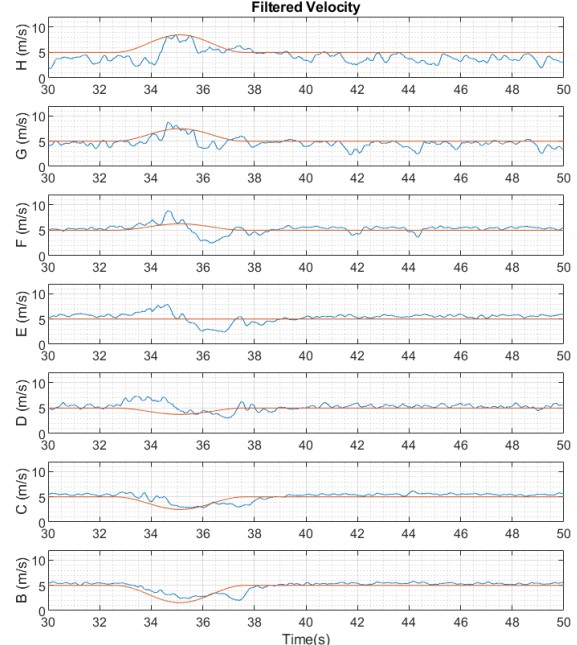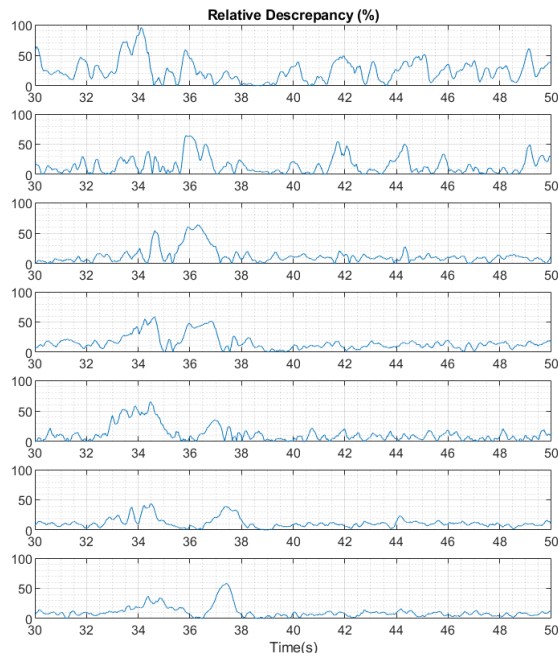

(a) EVS

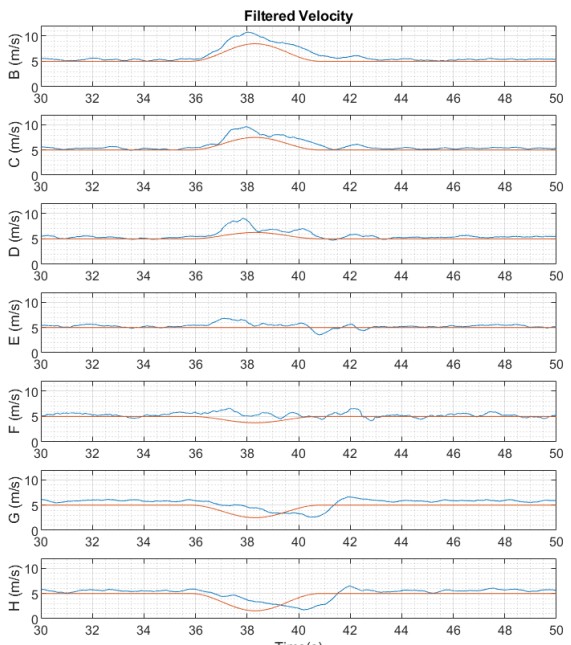
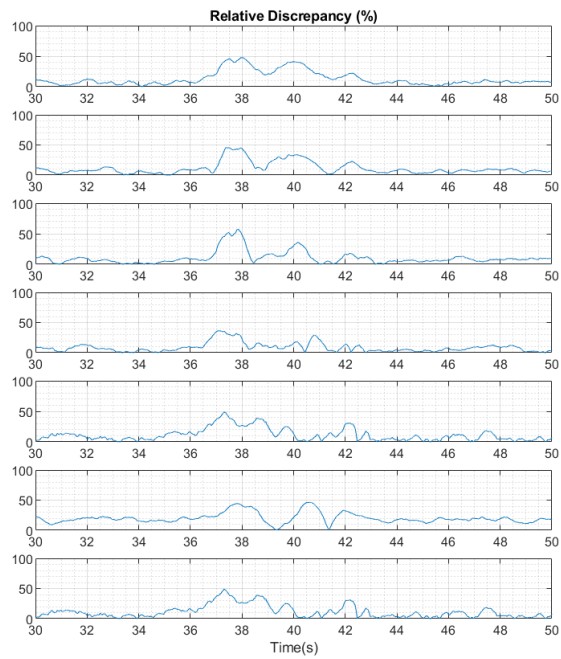

(b) EHS

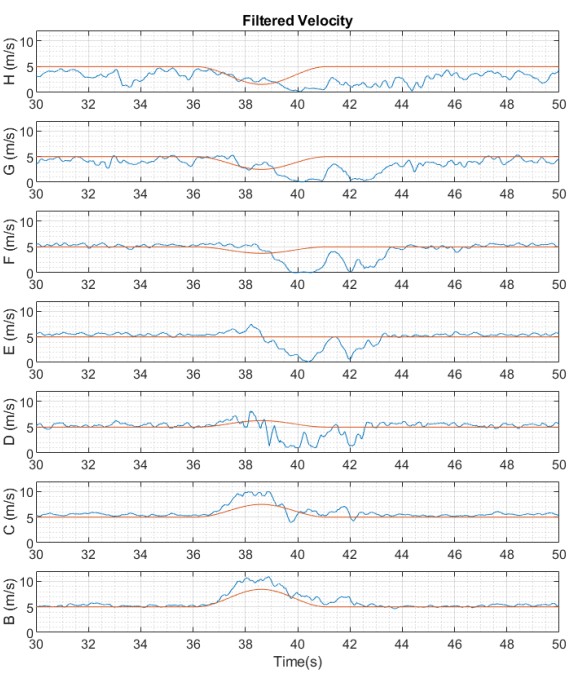
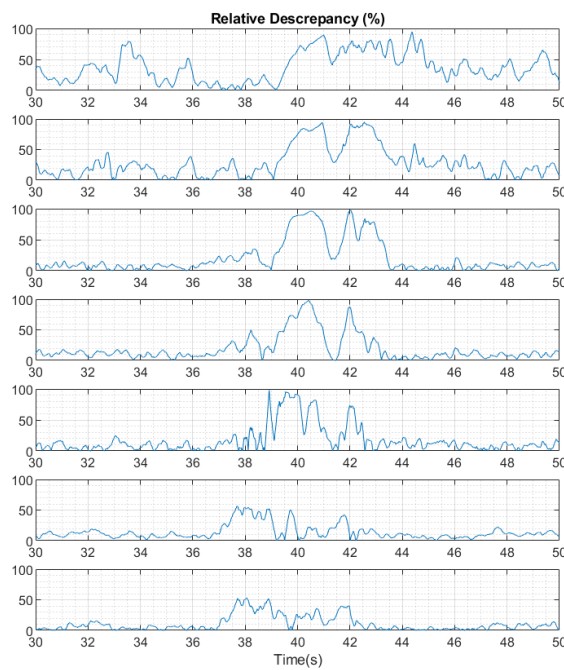

(c) negative EVS

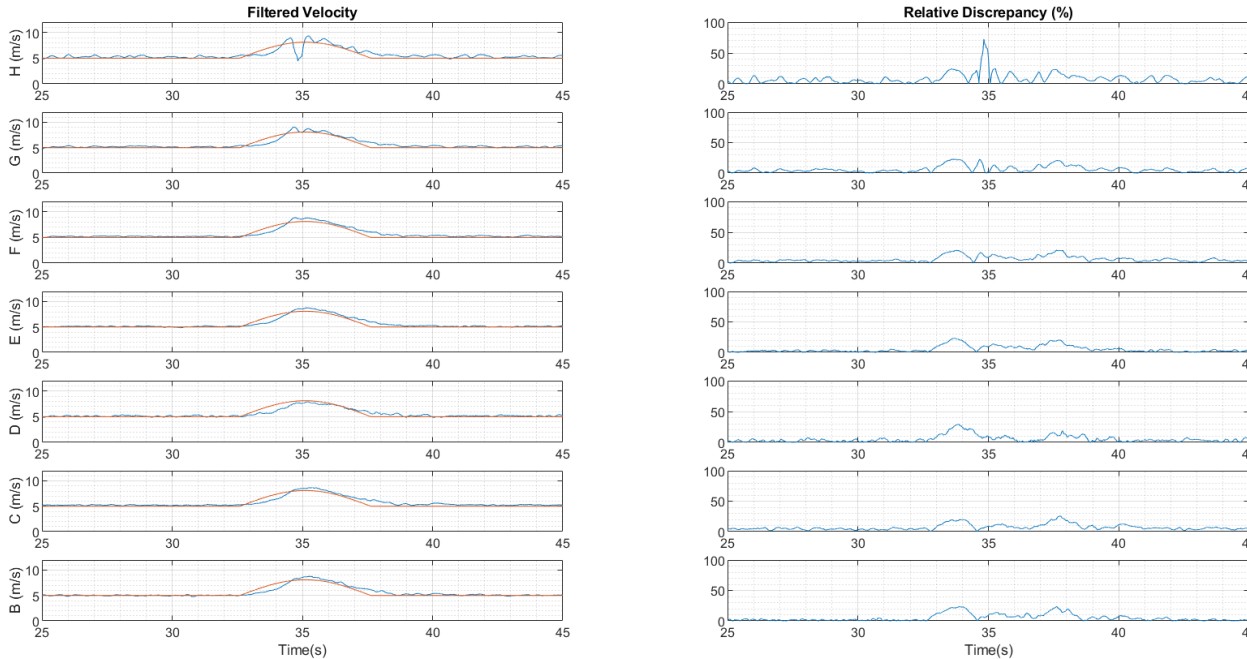

(d) EOG generated with changing fan powers, vertical measurements

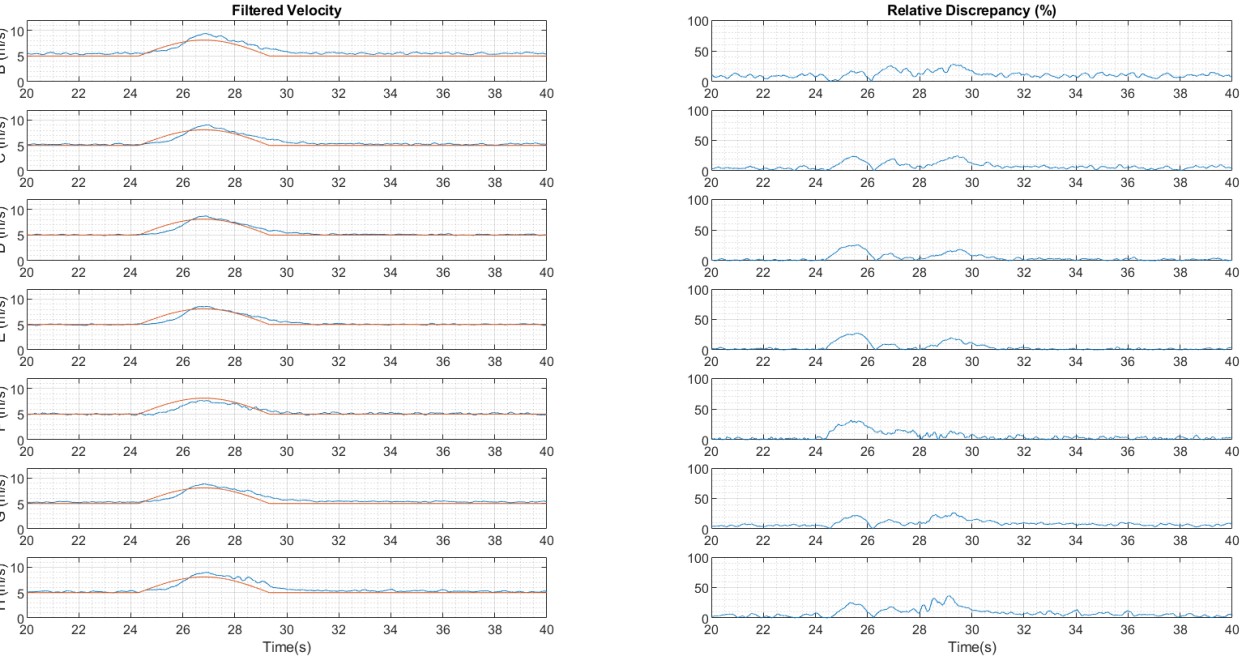

(e) EOG generated with changing fan powers, horizontal measurements

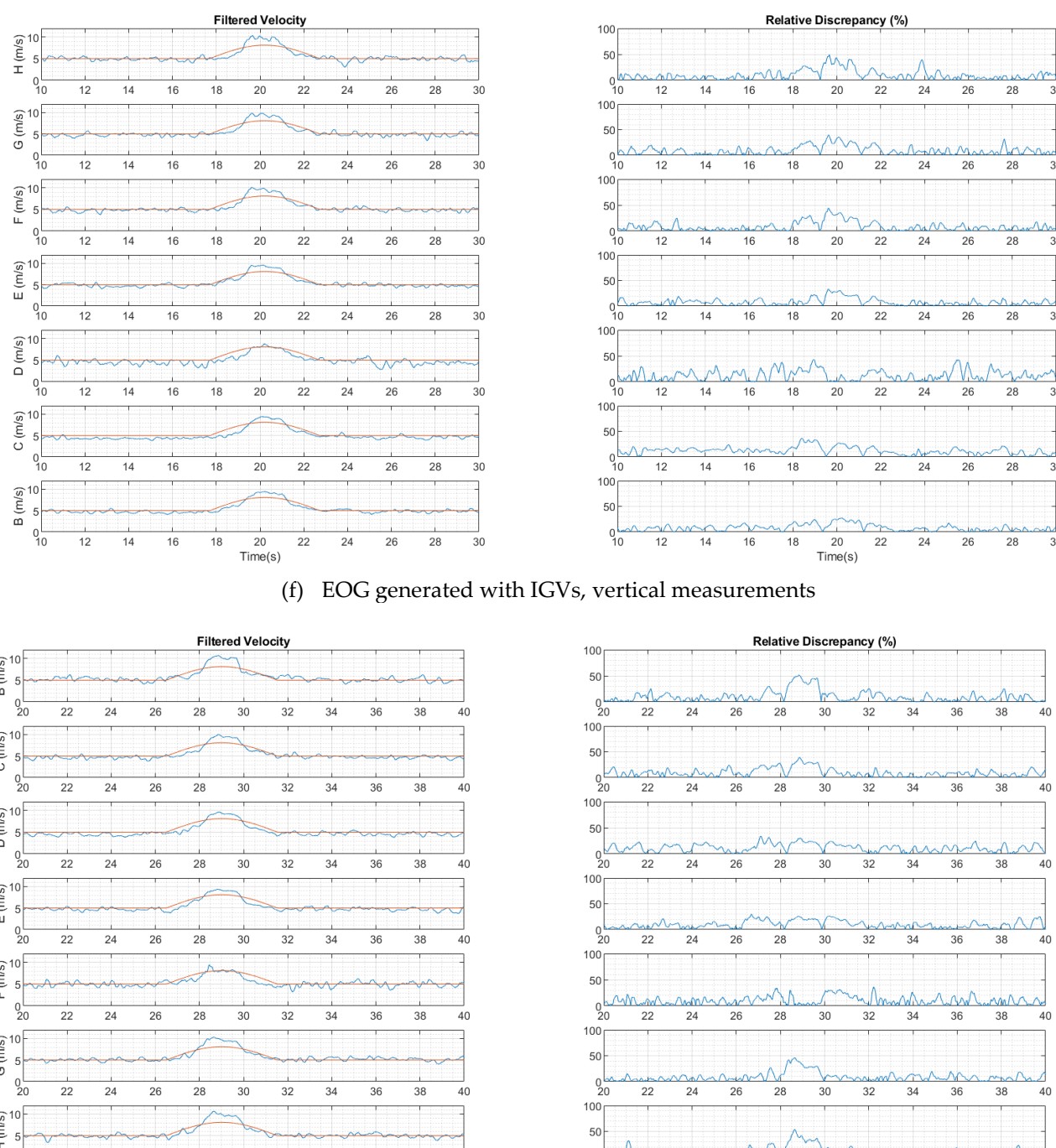

(f) EOG generated with IGVs, vertical measurements

(g) EOG generated with IGVs, horizontal measurements

**Figure 12. Filtred velocity time history at each probe (with the layout presented in Figure 6) as blue solid-line compared with prescribed extreme event velocity as the orange solid-line (left columns), time history of relative instantaneous velocity discrepancy normalized by average velocity (right columns)**

The most consistent EOG was generated by using IGVs in terms of uniformity, symmetry and peak factor at the test section (Figure 12f & g). The only noticeable inconsistency of this simulated EOG is due to effect of the contraction walls which resulted in slightly higher velocity peaks in probe H and B in Figure 12g that are in 1.1 m offset from center. The generated gust also has positive curvature on the both rising and falling sides. Even though, the simplified target gust has a negative curvature.

### 4. Conclusion

A hybrid experimental/numerical study has been carried out to investigate the possibility of creating extreme conditions for a scaled HAWT based on the IEC 61400-1 standard, in particular the EOG and EWSs, using a unique 60 fan setup in the WindEEE dome at Western University. These conditions were tailored for a 2.2 m diameter test HAWT with the aim to relate this work to full-scale wind turbines. Therefore, a length and time scaling approach based on tip vortex propagation in the wake was introduced. The resulting time scale is a function of the free stream velocity, tip speed ratio and diameter of the rotor.

A simplified numerical model was first developed and tuned based on a set of steady ABL flow data; the model used a simplified geometry of the WindEEE testing chamber and did not simulate the flow recirculation in the outer shell. The model also treated the fans simply as velocity inlet boundary conditions with the same efficiencies. Yet, it gave a good understanding of the relation between fan power set-points and the flow field at relevant part of the test chamber, which then was used to predict the fan setups for the physical simulation of the extreme events. For future target scenarios the numerical model can be useful to obtain the primary setup, however field adjustments are recommended.

Steady experiments corresponding to the peak of the shear cases showed that the fans act non-linearly and have different individual efficiencies, especially the top and bottom rows due to the sharp recirculation angle at the suction side of the 60 fan wall. This has not been taken into account in the simplified CFD model and consequently resulted in discrepancies between experiments and the standard shear. By quantifying these discrepancies, corrections can be applied to improve the replication of these events. The unsteady shear flow experiments showed that even though the desired peak factor was generated the high and low velocity peaks reach the test section with a time lag. This can be corrected in the future by providing a phase difference in actuations between the top and bottom rows of fans.

In generation of the EOG by dynamic change of the fan powers, the flow field was more consistent than the EWS compared to their own baselines; the combination of 60 operating fans and the contraction walls helped unifying the flow field. Yet, in fast power transitions the flow field showed some unpredictability and inconsistency close to the ceiling of the test chamber. Generating uniform gusts using the IGVs produced the best results in terms of time scale and peak factor, as well as

flow field uniformity and reproducibility. Considering the simplified gust profile without the velocity drops, the generated gust imitates the simplified theoretical profile.

Overall, this study demonstrated promising results using a hybrid numerical/experimental approach for the simulation of extreme wind conditions. These extreme gust conditions can be used with minor modifications in future physical tests to investigate their effects on different aspects of wind turbines' performances. Furthermore, a detail investigation into the

reproducibility of these extreme events, specifically the cases generated by dynamic change of the fan powers, is recommended.

**Authors contributions**

KS developed the numerical model and the scaling with direct supervision from CC. KS carried out all the experiments with supervision of HH. KS wrote the main body of the paper with input from all authors.

**Competing interests**

The authors declare that they have no conflict of interest

**Acknowledgements**

All authors thank Gerald Dafoe and Tristan Cormier for helping with the measurement setups. The present work is supported by the UWO, IESVic and the NSERC.

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
