# Peer review of "Experimental and Numerical Simulation of Extreme Operational Conditions for Horizontal Axis Wind Turbines Based on the IEC Standard"

_Wind Energy Science, 2020_

## Referee Comment (RC1) · Anonymous Referee #1 · 29 May 2020

[referee-annotated manuscript omitted]

---

## Referee Comment (RC2) · Anonymous Referee #2 · 2 Jun 2020

This paper aims to show a process chain to generate gusts and shear flows in the WindEEE Dome.

In the introduction, the problematic of gusts and shear flows in wind energy application are discussed and the measures proposed by the IEC 61400-1 are introduced. Next, the process chain is presented by generating a numerical model of the setup, validating the model and introducing the experimental setup. Then, the results are presented. First, the static shear flow results are shown, and then, the dynamic results are presented by explaining the numerical procedure to obtain the generator settings

and comparing the experimental results to the IEC standard.

Overall, this study can be seen as a first proof of concept of the capability and the limits of the WindEEE Dome to generate gusts and shear flows. The possibilities of flow generation with the WindEEE Dome are highly interesting and valuable for experiments. However, this study faces several issues that need to be addressed before publication. The introduction lacks motivation/comments on why doing experiments and why doing a new setup as well as commenting on existing setups and recent developments. There is also no comment on how simulations can/will be used to complement this study.

Also, I am missing a clear objective of this study in the introduction. As this paper serves as a proof of concept, this should be emphasized.

The Methodology lacks structure and it should be pointed out clearly that the idea of this study is to

a) generate a test chain where a new numerical model of the WindEEE Dome can be used to predict fan settings so that inflow conditions can be modeled/prepared before the experiment

b) give a proof of concept of the capability of the WindEEE Dome to generate roughly IEC conform gusts/shears and the capability of the numerical model to predict fan settings correctly

In the Results section, the mean velocity evolution of the three extreme wind cases (extreme operating gust and extreme vertical/horizontal shear) is presented. It is shown that properties roughly similar to the IEC recommendations can be achieved. For experiments in the future, I would recommend to show the reproducibility of the flows.

There are many points in the paper that need addressing, and they can be found in the following. Also, the paper needs language editing.

**Grammar/language**
l. 3 "WERE operated"
ll. 14: this sentence is quite long and the second part needs rephrasing
l 49 "smaller gusts . . . cause" (not causes)
l. 95: analyze not analysis
l. 125 "Another capability at the inlet wall utilizes fans with adjustable. . ." => another feature are the fans with adjustable. . ..
l. 146 parameterS
l. 203 time scale
you are writing XXm/s (no space), XX m/s (space) and XX seconds - please use equal convention for all cases
l. 243 "that ARE examined"
l. 244 "Using the tuned numerical model setups, the V-c and H-c domains were used"
l. 263 "resulting in a . . ." (not "an")

**Abstract**
The goal of this work as well as some results should be pointed out more clearly. Currently, the description of the WindEEE Dome is dominant, and while the setup is very impressive, it might be good to keep the details to the setup section. This will shift the focus towards the study that was performed.

ll. 19: HAWT is not introduced while all other abbreviations are introduced, in addition, as HAWT stands for horizontal axis wind turbine, it reads " horizontal axis wind turbine scaled turbine"

**Introduction**
l. 27 "low carbon energy" => "renewable energies" ?
l. 29 "life time"

ll. 31 "Therefore, the design of wind energy systems must consider extreme environmental conditions with statistically accurate return periods."
It was shown that extreme conditions occur more often in the ABL than predicted by a Gaussian distribution by Wächter et al. (2012), however, the statistics used in the IEC standard for the prediction of the probability of extreme events are Gaussian statistics. Matthias Wächter, Hendrik Heißelmann, Michael Hölling, Allan Morales, Patrick Milan, Tanja Muİĺcke, Joachim Peinke, Nico Reinke  Philip Rinn (2012): The turbulent nature of the atmospheric boundary layer and its impact on the wind energy conversion process, Journal of Turbulence, 13, N26

ll. 35 The 50 year return period is both noted in the third and fourth edition of the IEC 61400-1: "For the steady extreme wind model, the extreme wind speed, Ve50, with a recurrence period of 50 years, and the extreme wind speed, Ve1, with a recurrence period of 1 year, shall be computed as a function of height, z, using the following equations:…" [IEC-61400-1 ed. 3, p. 25ff]
For the EOG, a return period of 50 years is actually named.
Further, none of the four editions of the IEC 61400-1 standard appears in your sources even though you are using formulas and name all existing editions of the standard.

I can not follow your argumentation why the second version of the IEC standard should be chosen over the third (or fourth, depending on when the experiments were carried out) since the main difference of the EOG is that $\beta = 3.3$.

ll. 39 While steady state wind tunnel tests may be more common, dynamic tests have been carried out in aerodynamics for a long time and in wind energy research, the setups have improved a lot over the past years, too (see comment to ll. 96 for some literature)

l. 50 What is the "gust slicing effect"?

ll. 50 "These small gusts also can cause instabilities in the power output of wind turbine generators. For a small electricity network, these instabilities in power generation can cause serious problems for managing power transmission and distribution."
A source is needed for this statement.
What about the impact of large gusts on the power output and grid stability?

ll. 52 "The worst case is when a peak gust wind speed is higher than the wind turbine cut-out speed, which if prolonged enough can cause the control system to abruptly stop the wind turbine (Hansen, 2015)."
I assume, that "worst case" refers here to the high loads that this situation results in, however, before, you are writing about the stability of the grid which may be confusing.

ll. 54 "The most reasonable solution would be an adjustable generator load or adjusting blade pitch angles after detection of the gust"
This sentence is a bit misleading as a system must be in place in order to detect the wind field approaching the rotor. You could rephrase the sentence, e.g. "Ideally, the wind field approaching the turbine is measured so that the turbine control can adjust the generator load or blade pitch accordingly"
Could you also please comment on whether a control system with pre-monitoring would be sufficiently fast to react to EOGs?

ll. 64 "If extreme enough, the blades may experience a phenomenon known as dynamic stall (Hansen, 2015;Gharali and Johnson, 2015). All these together will result in instability and reduction in power generation, as well as highly dynamic fatigue loads on the system"
dynamic stall may also occur in case of the EOG. What do you mean by "instability in power generation"? Maybe, "high fluctuation in power generation" would be more accurate?

ll. 67 "These transient shears can happen for similar reasons as uniform gusts, but

mostly happen within wind farms, where the wind turbines sometimes operated in the wakes of other operating up-stream wind turbines"
With respect to shear, the main problem is that downstream turbines may partially be exposed to the wakes of upstream turbines.

ll. 71 "This standard also defines a classification for commercial wind turbines based on a reference wind speed and turbulence intensity, in a way that covers most on-shore applications. The Turbulence Intensity (TI) is defined as the ratio of the standard deviation of wind speed fluctuations to the mean 10 min averaged wind speed. TI levels of16Please cite the edition of the IEC 61400-1 standard. While in the top, you are claiming to use the gust definition of the second edition, these wind turbine classes are from the third edition (and they have been expanded in the fourth edition).
Please specify, that in the definition of the turbulence intensity, both the standard deviation and the mean velocity are calculated for the respective 10min interval.

ll. 78 $I_{ref}$, $t$, $\sigma_u$, $\bar{V}$,$Z$ and $Z_{hub}$ are not introduced

ll. 96 "Experimental investigations are typically limited to steady state conditions (Snel et al., 2007; Sørensen et al., 2002)."
The introduction is generally lacking an overview of devices capable of generating unsteady flows that have been used in the past also for the generation of gusts. I also think that a broader overview on how wind tunnel experiments address certain turbulent aspects to have a more realistic setup should be mentioned. Some points that could be addressed (The citations in the following are meant as help to give an idea of what has been done, the are meant as a suggested starting point which does not mean all need to be cited):

- There are different kinds of steady state experiments: Some are carried out in uniform laminar or turbulent inflow and some are carried out in boundary layer flows.

- The active grid has been used to generate turbulent inflow conditions that have more dynamic and higher variation (e.g. Jin (2016) Effects of Freestream Turbulence in a Model Wind Turbine Wake, Li (2020) The near field of a lab-scale wind turbine in tailored turbulent shear flows, Rockel (2017) Dynamik wake development of a floating turbine in free pitch motion subjected to turbulent inflow generated with an active grid)

- Aerodynamic experiments have been carried out in sinusoidal flows/using pitching and/or plunging airfoils to investigate dynamic stall which is also found on wind turbines (setups for sinusoidal flows: e.g. Tang (1996), Wei (2019), experiments on airfoil: e.g. Choudhry (2014) An insight into the dynamic stall lift characteristics (and sources)).

- There have been several experiments in unsteady conditions, often generated by active grids, both for whole rotors and for airfoil investigations:
Petrović et al (2019) Wind tunnel setup for experimental validation of wind turbine control concepts under tailor-made reproducible wind conditions
Wester (2018) High Speed PIV measurements of an adaptive camber airfoil under highly gusty inflow conditions
Schottler (2017) On the impact of non-Gaussian wind statistics on wind turbines – an experimental approach
Neunaber et al (2017) Comparison of the Development of a Wind Turbine Wake Under Different Inflow Conditions
Sakar and Haan (2008) Design and testing of Iowa State University's AABL Wind and Gust Tunnel
Neunaber and Braud (2020) Characterization of a new perturbation system for gust generation: The Chopper

ll. 97 "Only a limited number of studies have looked at transitory or turbulent wind conditions (Peinke et al.)"

The citation does not give information on the year and the journal. Also, the authors are not cited in correct order. It appears to be a proceeding paper from the EWEC 2004 and is therefore again not state of the art.

**Methodology WindEEE Dome**

p. 5, fig. 2 the text is not readable in A4 format, ti is too small

ll. 111 "A render of the inner shell of the test chamber for straight flows is shown in Figure 2a. The test chamber is in turn surrounded by an outer shell. It has a total 106 fans, including 60 fans installed on one wall and 40 fans over the other five peripheral walls. There are also 6 larger fans in a plenum above the test chamber."
The readability could be improved

ll. 115 "To simulate a straight flow, the louvers at the top and peripheral sides of the test section are closed and the flow goes from the 60 fans to the center of the test section and then through the mesh of the wall at the opposite end, before passing through heat exchangers and recirculating over the top, back to the 60 fans' inlet."
The louvers at the top have not been mentioned before.
Is figure 2b showing the "normal wind tunnel" configuration? Does the air circulate as shown in figure 2a, or 2b, or both in this configuration? Are the 6 fans in the plenum used in the straight configuration to reach higher velocities?

ll. 121 "The power set-points of the 60 fans can be adjusted by a spreadsheet file with 60 columns. The numbering of the fans starts with the top left fan, row by row ending with the bottom right fan in Figure 2. The operating software at the facility can read the spread sheet file and switch power set-points as fast as 2Hz."
I do not think that mentioning the organization of fan control data in spreadsheets adds necessary information, you could also write "The power set-points of the 60 fans can be adjusted by the software as fast as 2Hz"

l. 124 "due to rotational inertia of the fans and electrical current filtering it takes 3 s"

Does this mean that it takes 3s for the flow to adjust or 3s for the fans to adjust?

**Numerical Flow Analysis Setup and Tuning/Validation**

This part is missing some structure that helps the reader understand the procedure (this is how I understand it):

1. you want to have a "numerical setup" so that you have some guidance on how to set the fan power and how to optimize the simulation

   - you define a domain
   - you generate different meshes
   - you run the simulation and validate it with experimental data, then you optimize the simulation and finally you know which mesh is suitable while sufficiently accurate

2. you use the optimized simulation to generate a look-up table of the fan power

table 1: The caption could be more precise

l. 145 it would be helpful to mention that you have 3 meshes (and probably why you have 3 test cases)

l. 152 referencing to table 1 or introducing V and V-c would improve readability

ll. 155: Where are the 13/31 m/s found? In the center of the test section? Are these the data from experiments? Does that mean that the inlet fan velocity in the simulation corresponds to 16.5 m/s in order to reach the 13/31 m/s in the experiments?

l. 168 this is the first time that you mention that an ABL profile is aimed for

figure 4: where are these profiles measured/simulated?

**Experimental setup for velocity measurements**

ll. 190 "..to measure three velocity components with measuring range from 2 to 45 m/s with $\pm$0.5 m/s accuracy (Turbulent Flow Instrumentation Pty Ltd). In this study the average wind velocity was 5 m/s, therefore all of the wind measurements have 10The source is not appearing in your references.
You are mixing accuracy and uncertainty which are two different things.

l. 197/fig. 7: As you show the turbine in fig. 7, I would recommend rephrasing this sentence to really emphasize that the turbine is NOT used in these experiments.

**Gust length and time scaling**
ll. 203 please comment on why you are choosing 4 loops of a blade tip vortex

l. 206 $L = 4 \cdot L'$ is missing as definition (T as well)

ll. 220 this paragraph is confusing:

- do I understand correctly, that TSR = 1.1 and u = 5m/s are the working conditions of your model wind turbine, and that these conditions are chosen to match the maximum speed at which your gust can be generated due to the inertia of the system? If so, why do you do the argumentation of the gust scaling if in the end, you will alter the TSR because you cannot generate rapid gusts? I understand this paper as proof of concept and since you are generating the shortest gust possible, the argumentation is probably better placed in a paper where you are actually using the turbine.

- Why are you commenting on how to improve the Re mismatch? Are you planning to do this? Can you actually change the fluid in the WindEEE Dome? You could comment on the consequences of low Re for your experiment as an alternative.

- l. 228 "In this setup, the ratio of the length and time scale become 5.23 and 2.61 respectively." I would place this sentence in l. 223 or 224

ll. 234 you should emphasize that you use the new gust duration T instead of the IEC gust duration T

ll. 235 could you give a citation where this gust has been simplified? Also, it should be added that the important part is the amplitude by which the wind increases (here from 3.5 m/s to 9.5 m/s in 1.25 s (2.4 s IEC) ) - the drop should therefore not simply be ignored

**Results and discussion**

figure 10 Do I see correctly that only the central 20 fans are used when the contraction nozzle is installed?
=> found information in l. 268
=> Did you only use the central 20 fans when doing the simulations (e.g. tab. 3)?

l. 252 / figure 11 "Figure 11 shows the average velocity at each probe including the range of standard deviation"
what do you mean by "range of standard deviation"?
if I interpret the figures correctly, you have turbulence intensities of up to 50% (fig. 11 c, 1.5m height: $V \approx 5m/s$, $\sigma \approx 2.5m/s$. You argue that this is due to vortex forming. Could you please comment on possible consequences during your experiment?

ll. 257 "The largest error exists in the horizontal shear case.."
If you are referring to the standard deviation with "errors", I would like to point out that the standard deviation of a turbulent flow is not suitable to serve as an error as the fluctuations are inherent to the flow and the standard deviation gives an idea of the strength of the fluctuations.

figure 11 d / equation (8) "relative difference" is more accurate than "relative error" for describing the difference between the measurements and the IEC standard

**Unsteady experiments**

For the EOG, have all fans be operated?

ll. 267 Again, I don't see a point in mentioning the spreadsheet - the information that all fans except for two rows were run at 39

l. 271 "Therefore, the file has many rows of 39Above, you are stating that only 5 columns of fans are used, I would imagine that therefore, 40 out of 60 fans are not in use?

ll. 277 "For example if we put the fan power setups for the extreme condition at the 11th and 12th row in the file, the event start happening"
As a reader, I have no idea, what fan row 11 and 12 of the spreadsheet correspond to, and it is also not important. This is not a manual on how to run the wind tunnel but a presentation of the setup and the results.

figure 12

- the writing is too small

- is this a phase average? Did you verify the reproducibility of the gust?

- it appears that one cobra probe does measure a significantly lower velocity (more than 0.5 m/s you give as probe accuracy) than the other sensors: There is a bump in fig. 12 d-g. Since both vertical and horizontal measurements have been performed, this appears to be a problem with the probe rather than an alteration in the flow field (probe F for hor. measurements/ D for vert. measurements - but since F and D are symmetrically ordered around the center, it might actually be the same probe with different labelling?). Did you make sure the cobra probes are calibrated the same way or test the measured velocity variation between the probes?

figure 13

- it would be helpful to remind the reader that B-H are the probe names and that positions can be found in figure xx

- you could comment on the fluctuations that are visible in fig. 13a and c since they are increasing with increasing height of the sensor

ll. 300 "Based on Figure 13d for the EOG generated with changing fan powers, the velocity at the upper height in the test section is not achieving a totally uniform flow condition(time series from probe H)."
Did you check how the raw data looks? Considering the rather broad moving average window of 0.2s / 400 data points, the "hole" at t = 35s might stem from some not collected data points which may for cobra probes occur if the flow leaves the measurement area (too high/low velocity/ too large flow angle)

ll. 301 "However, the desired peak factor has been generated"
while the ration $V/V_{max}$ may be similar to the IEC EOG, you do not achieve the amplitude. Also, a comment on the rise and fall time as compared to the IEC EOG would be interesting.

**Conclusions**
ll. 327 "By ignoring the sudden velocity drops in the theoretical gust profile, the generated gusts would become identical to the standard." I disagree because your amplitude is lower while the rise and fall times are higher.

---

## Author Response (AR1)

**RC2 comments,**

**Introduction**

- Could you also please comment on whether a control system with pre-monitoring would be sufficiently fast to react to EOGs?
- Ans: A LIDAR beam can measure the approaching flow field of the size of the rotor diameter at 100-150 m upstream. Based on the average wind velocities it gives about 8 to 15 seconds to wind turbine to react with collective pitch or individual pitch control. But still LIDAR technology has limitations and it needs more developments, but it is very promising [Bossanyi, E. A., Kumar, A. and Hugues-Salas, O.: Wind turbine control applications of turbine-mounted LIDAR, J. Phys. Conf. Ser., 555, 012011, doi:10.1088/1742-6596/555/1/012011, 2014.]

**Gust length and time scaling**

- please comment on why you are choosing 4 loops of a blade tip vortex?
- Ans: the gust time in the IEC is between 10.5 or 15 seconds. The rated rotor speed of the commercial turbines based on their size in nominal operational condition is between about 12 to 20 RPM. Considering 16 RPM rotor speed and 13 s gust time in average, the rotor does 3-4 complete rotation during the gust time. This is the simplified idea behind our scaling consistent with the dynamic processes involved related to the progression downstream of trailed vorticity in the wake.
- could you give a citation where this gust has been simplified? Also, it should be added that the important part is the amplitude by which the wind increases (here from 3.5 m/s to 9.5 m/s in 1.25 s (2.4 s IEC)) the drop should therefore not simply be ignored.
- Ans: We acknowledge your comment, I have changed the words that have been used in this section, by mentioning that this is just the closest time duration that we can get with the current setup. The main goal was just to capture the amplitude from mean to the peak gust. The IEC standard gust includes an initial dip to match field experiments in which gusts are preceded by a lull; the present of the lull when the turbine is otherwise operating at mean conditions is likely benign, with the important gust impacts being realized from the extent of the wind speed excursion above the mean wind speed.

**Results**

- if I interpret the figures correctly, you have turbulence intensities of up to 50% (fig. 11 c, 1.5m height: V \_ 5m/s, \_ 2:5m/s. You argue that this is due to vortex forming. Could you please comment on possible consequences during your experiment?
- Ans: as you know velocity shear is the main element of turbulence production in the TKE transport equation. With this high shear strong vortexes form and increase the momentum mixing between different layers. This high amount of turbulence intensity definitely affects our results from the cobra probes. According to the manufacturer up to 30% of turbulence intensity the accuracy of measurements is about +-0.5 m/s in velocity range of 2- 40 m/s, however at higher turbulence the range of accuracy is unknown but certainly less accurate.
- Fig12, is this a phase average? Did you verify the reproducibility of the gust? it appears that one cobra probe does measure a significantly lower velocity (more than 0.5 m/s you give as probe

accuracy) than the other sensors: There is a bump in fig. 12 d-g. Since both vertical and horizontal measurements have been performed, this appears to be a problem with the probe rather than an alteration in the flow field (probe F for hor. measurements/ D for vert. measurements – but since F and D are symmetrically ordered around the center, it might actually be the same probe with different labelling?). Did you make sure the cobra probes are calibrated the same way or test the measured velocity variation between the probes?

- Ans: Yes, all the velocity figures are phased averaged with 0.2 s window. And you are right it is the same probe D recording consistently lower velocities than others. All of the probes have been calibrated identically. We noted this in the text.
- "Based on Figure 13d for the EOG generated with changing fan powers, the velocity at the upper height in the test section is not achieving a totally uniform flow condition (time series from probe H)." Did you check how the raw data looks? Considering the rather broad moving average window of 0.2s / 400 data points, the "hole" at t = 35s might stem from some not collected data points which may for cobra probes occur if the flow leaves the measurement area (too high/low velocity/ too large flow angle).
- Ans: In other experiments we saw the same inconsistency of the flow from the top row of the fans when we create uniform gusts. The velocity ranges and directions are in the cone shape measurement area of the probes.
- while the ration V=Vmax may be similar to the IEC EOG, you do not achieve the amplitude. Also, a comment on the rise and fall time as compared to the IEC EOG would be interesting.
- Ans: You are right, but we tried to develop the closest possible gust to the IEC. With the complex system of the fans, the velocity drops is not possible to replicate. But we tried to at least reproduce the same time duration for all EOG and EWSs events (they all have 5 seconds for rising and falling periods). We tried to change the words that we used in the text as well to reflect this.

**Conclusion**

• "By ignoring the sudden velocity drops in the theoretical gust profile, the generated gusts would become identical to the standard." I disagree because your amplitude is lower while the rise and fall times are higher.

• Ans: You are correct; as per last comment, we changed our language in the conclusion.

**RC1** Comments,**

**Abstract**

- suggesting restructuring the abstract
- Ans: We did, thank you.

**introduction**

- ("rather than trying to replicate the much more complex current standard".) This is not really a justification can the authors please comment?
- Ans: the current standard specifies a statistical approach to gusts; this project was a first step in wind tunnel experimentation to create a simplified gust/ shear based on a previously established industry standard. So, we chose IEC 3rd edition to do so, which has deterministic formula for these events. We have revised the text to clarify the context of the study, using a deterministic gust, relative to a stochastic handling of gust/extreme wind events which is more suitable for numerical simulations purposes but would be extremely difficult to achieve experimentally.

**Numerical analysis**

- (other parameter was left as default values) were these appropriate? Can the authors please comment?
- Ans: other parameters for automated mesh function in StarCCM are surface curvature, surface growth rate and mesh density. Usually the default for surface curvature is 6 degree which means it put a node on the surface at each 6 change of degree. The growth rate is 20% so then we do not have a big jump in size in proximity elements. The mesh density is usually being used for bluff bodies when you need to refine the wake of an object.

**Experimental setup**

- Not Enough details are included in the setup on the probes, justification of sampling frequency and time, a/d conversion, the measurement stations, etc.
- Ans: The sampling frequency was set to the highest (2k Hz) so we could have flexibility for our moving average filtering later. All of signals from the cobra probes were connected to a specific deck/ interface box and then a regular a/d conversion card then a windows laptop. The process is straight forward and accessible on their website. The text has been clarified in this respect https://www.turbulentflow.com.au/Downloads/Cat\_CobraProbe.pdf.
- (fig 7) How were these value chosen? Can the authors please comment?
- Ans: the values are based on a 2.2 m wind turbine with 1.9 m hub height that also is visible in this figure. The goal was to just have acceptable resolution of the flow field in the rotor swept area with lowest number of the cobra probes.

**scaling**

- I suggest considering collating figures 9 and 1.
- Ans: At first, we thought the same but then we determined it improves the readability of the paper by displaying two separate figures. Because our scaling method needs rather long explanation, we didn't want to confuse readers as they progress through the paper.

**Results**

- (largest error exist in horizontal shear) Can the authors speculate on why this is the case? Have any attempt been made to match the level of agreement seen in the vertical shear case?
- Ans: the CFD model is tuned based on previously tested ABL flows. Also, in CFD all the fans were considered identical with the same amount of efficiency by ignoring the flow recirculation. The values in these steady shears as well as unsteady cases in physical experiments were taken directly from CFD predictions so we have nonconformity with what we expected/ IEC. As we mentioned in the conclusion because of the complex nature of the fans and the geometry of WindEEE the developed CFD just should be used in preliminary stage but then field adjustment will be needed.

- (fig 13) this figure contains too many subfigures, which are not discussed in detail. I suggest condensing this information and perhaps report one plot as an example. The rest of the plots could be added as supplementary material (or similar).
- Ans: Thank you for your suggestion, we have added more discussion to the figures. Putting figures right along each other maybe at first glance looks confusing but it going to improve their readability once the reader understands the integrated and relative structure of the layout between subplots.

[revised manuscript text omitted]

$$= 5.6 \frac{m}{s},$$

Formatted Table

Formatted Table

Formatted Table

The  $V_{hub}U_{hub}$  is the average wind velocity at the at hub-height. The EOG with and b is a constant. Considering t as the instantaneous time and t = 0 at the beginning of the gust, the uniform EOG as function of time is defined as:

$$\begin{split} \mathcal{V}(t) &= \begin{cases} \frac{\overline{\mathcal{V}} - 0.37\beta \left(\frac{\sigma_{\overline{u}}}{1 + 0.1 \left(\frac{D}{A_{\overline{u}}}\right)}\right) \sin \frac{3\pi t}{T} \left(1 - \cos \frac{2\pi t}{T}\right); when \quad 0 \le t \le T, \\ \overline{\mathcal{V}}; \quad when \ t > T \ or \ t < 0, \end{cases} \tag{(2)} \\ &= \begin{cases} \frac{\overline{\mathcal{V}}_{hub}}{1 + 0.1 \left(\frac{D}{A_{u}}\right)}\right) \sin \frac{3\pi t}{T} \left(1 - \cos \frac{2\pi t}{T}\right); when \quad 0 \le t \le T, \\ \overline{\mathcal{V}}_{hub}; \quad when \ t > T \ or \ t < 0, \end{cases} \end{cases}$$

The factor  $\beta$  takes the value of 4.8 or 6.4 for gusts with recurrence periods of 1 or 50-years respectively. The duration of the gust T is specified as 10.5 s for 1-year and 14 s for 50-years return periods. The *D* is the diameter of the rotor, and  $\Lambda_u$  is the longitudinal turbulence scale parameter which is a function of the hub height:  $\frac{(Z_{hub})}{(Z_{hub})}$

$$A_{u} = \begin{cases} 0.7Z & for \ Z \le 60m_{r} \\ 42m & for \ Z > 60m_{r} \\ 42m & for \ Z_{hub} \le 60m_{r} \\ 42m & for \ Z_{hub} > 60m_{r} \end{cases}$$
(13)

The EVS and EHS have similar equations which can be added to or subtracted from the main uniform or ABL inflow. The EVS as function of height (Z) and time can be calculated using the Eq. (-4)(4):

$$\begin{aligned} & \frac{\mathcal{V}_{EVS}(Z,t)}{2} \\ &= \begin{cases} \left(\frac{Z-Z_{hub}}{D}\right) \left(2.5 + 0.2 \,\beta \sigma_u \left(\frac{D}{\Lambda_u}\right)^{0.25}\right) \left(1 - \cos\left(\frac{2\pi t}{T}\right)\right); when \ 0 \le t \le T, \\ 0 \ ; when \ t > T \ or \ t < 0, \end{cases} \end{aligned}$$

110

For a commercial B-III class HAWT with 92 m diameter rotor and 80 m hub height, at 10m10 m/s average velocity, the prescribed EOG and EVS for 1-year return period are presented at Figure 1. Figure 1. The time window in this figure starts and ends with the extreme event, which is 10.5s for 1-year return period. Generally speaking, the The peak factor of the EOG decreases with increasing size of the turbine or decreasing hub height, and vice versa for the EVS based on these equations.

115

These extreme models are relatively simple and are not able to capture the true coherent turbulent wind characteristics (Cheng and Bierbooms, 2001). This is the reason for recent editions of the IEC standard to utilize statistical methods for characterizing extreme gust event performance. This has been enabled by computational resources to analysis wind energy systems in dynamic wind environments. Experimental investigations are typically limited to steady state conditions (Snel et al., 2007; Sørensen et al., 2002). Only a limited number of studies have looked at transitory or turbulent wind conditions (Peinke et al.). Developing a method to experimentally test extreme conditions on rotors is a valuable contribution to researchers in this field, and as such is the main motivation and focus of this study examining deterministic transitory gust profile generation.

125 This Along with more common steady state experiments (Snel et al., 2007; Sørensen et al., 2002), developing transitory flow field experiments have attracted the interests of researchers during the past few decades (Lancelot et al., 2017; Ricci et al., 2017) to evaluate the various computational techniques or to directly investigate complex phenomena in different applications. To the authors' knowledge, in the wind energy field some efforts have been made to produce gusts using active

Formatted Table

grids (Petrović et al., 2019; Wester et al., 2018) and a chopper mechanism (Neunaber and Braud, 2020). Developing these
unsteady flow fields basically comes down to the experiment targets and the available wind tunnel facilities. In this study, the generation of the EOG and the EWSs unsteady flow fields with proper scaling (customised for a 2.2 m scaled HAWT) using 60 individually controlled jet fans in the WindEEE dome are considered. This work presents a new numerical model of the WindEEE dome test chamber which can be used to predict fan settings for any custom steady or unsteady 2D flow fields before the physical experiment, and the capability of this facility to physically generate the gusts and shears similar to IEC standard
during experiments. The focus of this paper is just on the time evolution of the simulated extreme conditions' flow fields which is a prologue for future experiments including an actual HAWT model.

The paper is organized in three sections beside the introduction and it is as follows. Section 22 details the development of the numerical model for the WindEEE test chamber which was used to obtain the fan setups to use in physical simulation of the gusts. This section also provides a length and time scaling of the gust which based on that the target gusts for
 experimental campaign are introduced. Section 33 presents the results from velocity measurements at the test section in two parts, firstly the steady shears to double checkassess the accuracy of the developed numerical model to simulate the shear layers and secondly the final transient simulated gusts and their comparison with IEC standard. Section 44 provides the some conclusions.

**2. Methodology**

145

**2.1. WindEEE Dome**

The physical experiments were conducted in the WindEEE Dome at Western University, Canada. This is a versatile facility withthat can be run at different modes for creating various three dimensional and non-stationary wind systems (Hangan et al., 2017). It has an inner test chamber with a 25 m diameter hexagonal footprint and 3.8 m height. A render of the The dome inner shell of and the flow path in the test chamber for straightclosed-circuit 2D flow mode (e.g. ABL, shear flows is shownand etc) are rendered in Figure 2Figure 2a. The test chamber is in turn surrounded by an outer shell. It has a total 106 fans, including 60 fans installed on one wall and 40 fans over the other five peripheral walls. There are also 6 larger fans in a plenum above the test chamber. A which are mostly used for generating 3D flows like tornados and downbursts. The side view schematic of the WindEEE Domedome is shown at Figure 2Figure 2b. In general, to describe the arrangement of multiple fans allows for sheared, yawed and circulating flows to be created flow recirculation path in the dome. To simulate a straight flowclosed-circuit

155 2D flow mode. In this mode, the louvers at the top and peripheral sides of the test section are closed and the flow goes fromenergizes only by the 60 fans then reaches to the test section (center of the test sectionchamber) and then exits the test chamber through the mesh of the wall at the opposite end, before then recirculating over the top while passing through the heat exchangers, and recirculating over the top, finally back to the 60 fans' inlet. Each fan is 0.8 m in diameter with 30 kW nominal maximum power. In order to reach higher velocities and better flow uniformity characteristics at the center of the test chamber,

160 a two-dimensional contraction can be setup to streamline the flow as shown in Figure 2c.

Field Code Changed

---

## Referee Report (RR1)

In this second version of the paper "Numerical and Experimental Simulation of Extreme Operational Conditions for Horizontal Axis Wind Turbines Based on the IEC Standard", the authors worked on the presentation of the results and overall, the corrections improved the readability. However, I do feel that some of my remarks have not been addressed sufficiently scientifically - both in the paper and in the author's reply. In addition, there are some open questions.
Also, the paper needs language editing.
The following major corrections should be done and answers to these questions should be given:

**1 Introduction**

When bringing up the IEC 61400-1 standard, I have the impression that there is some confusion at the authors' side regarding the versions, see comments in the following:

- From my understanding, the EOG and the EWS are not thrown out in the 4th edition but the standard has been expanded. This should be clarified.

- The wind speed information you are giving in l. 74 is not accurate. If you are bringing up the specific values, you should explain what the values you are just citing here actually mean - not a normal inflow velocity of 50 m/s but the reference wind speed (for the definition, please look into IEC-61400-1 4th edition)

- While you are now citing the 3rd and 4th edition of the IEC 61400-1 standard, you are using the formulas of the 2nd version (at least, I suppose you do since you did not change the formulas). Change the formulas or the citation. Do not use wrong citations.

Also, the authors should point our why the gust slicing effect is a problem (recurring high loads while the turbine blade crosses the gust several times)

**2 Methodology**

**2.1 WindEEE Dome**

- in the text, both figure 2a and figure 2b are said to show "the closed-circuit 2D flow mode" while the caption says "render of test chamber and flow path" (a) and "side view schematic of the WindEEE Dome with flow path in closed-circuit straight flow mode" (b). Please clarify the description. Do I interpret the figures correctly when saying that the flow returns around the inner chamber (i.e. left, right, above)?

- l. 134: you are mentioning the repeatability of gusts and shear flows. I would like to know whether this has been shown somewhere and if a citation could be included.

**2.2 Numerical Flow Analysis Setup and Tuning/Validation**

In this part, you should be clear about using *one static* condition to validate the numerical setup while the numerical setup will afterwards be used dynamically to generate the gusts and the shear. This could also be used in the discussion of the deviation in the flow field due to the different performance in the upper fan rows (as you write, "By knowing these discrepancies, corrections can be applied to the fan inputs").

**2.3 gust length and time scaling**

- I could not find any reference of the 14s duration of the 50 year EOG in the IEC 61400-1 ed 3 and 4 standards, could you please give me the section where I can find this?

- l. 227 (question out of interest): with a TSR of 1.1 and an inflow velocity of 5 m/s, the rotational frequency of the turbine (R = 1.1m) would be less than 1Hz which appears very slow to me. Is there a mistake?

- ll. 235: It is not clear what you want to express here. Please rephrase this.

**3 Results and discussion**

**3.1 Steady wind shear**

- I would change the wording in line 263: you are talking about velocity deviations but the standard deviation gives you a measure of the fluctuations in the flow

- l. 279: it is not clear what the setting of 1.6 s is and what it implies. If the information is relevant, please be more specific

- ll. 311 I do not agree with the use of the term "intermittency" in this context because in turbulence, it is normally used for effects occurring in the flow *due* to the turbulence while here, in contrast, the short, strong dip in the velocity is not caused by turbulence but more likely due to the setup. I suggest rephrasing the explanation.

- A thought regarding the velocity "dip": When looking at the time series, it is obvious that the higher measurement positions have higher turbulence levels and in addition, probe H measures this "dip" - could this be explained by the upper fans having a less stable air supply than the lower fans who will suck more air from the sides and above? (due to the sharp angle between the upper row of the fans and the plenum, I would expect more air to flow from there to the lover fans) And could this dip be related (lack of air when suddenly increasing the velocity)?

**4 Conclusion**

- While you are talking about a "numerical and experimental study" both in your title and the discussion, I do not feel that this is an accurate description since you are describing how you build a numerical "setup-preparation chain" that you validate against previous ABL data sets, and this numerical setup is in the following used to set the vans. However, you are not showing the simulation results of the gusts and shears, and therefore it is not directly a numerical study in my opinion. What you compare are the experimental results and the expected results from the formulas. I therefore think it would aid if you would rephrase the first sentence of the conclusion.
  Is a numerical study with a simulation and experimental campaign for example with the turbine planned in the future?

- "The steady experiment runs corresponding to the peak of the shear cases show the fans act non-linearly and they have different individual efficiencies, especially the top and bottom rows despite our simplified assumptions for developing the CFD model." I would suggest rephrasing:
  "The steady experiment runs corresponding to the peak of the shear cases show THAT the fans act non-linearly and THAT they have different individual efficiencies, especially the top and bottom rows. This has not been taken into account in our simplified assumptions for developing the CFD model."

**5 Author Comments**

It would have been nice if specific information (for example the lines) on the changes that have been made in response to the reviewers' comments would have been added.

- When asking about a) the phase average of your gust and b) the repeatability, comment on fig. 12, I wanted to know whether you repeated the experiment several times and checked a) whether the gust events are similar and b) whether the plots are presenting an average over multiple gusts. I did understand that you use a filtering with a 0.2s moving average but this is not a phase average. Could you please clarify the above-mentioned points and add this in the paper?
  I could not find the information that the cobra probes show variations in the measurement of the mean velocity in the revised paper.

---

## Referee Report (RR2)

**Review of 'Numerical and Experimental Simulation of Extreme Operational Conditions for Horizontal Axis Wind Turbines Based on the IEC Standard' (MS No.: WES-2020-76.R1)**

by Kamran Shirzadeh et al.

The revised manuscript has largely been improved in its content, however, the technical standard is, in my opinion, still lacking in places. The language will also need a throughout proofread (e.g. see my comments on pages 1-4). I am generally satisfied with the content of this proposed review, and I would, therefore, recommend the manuscript's publication in WES once its presentation is improved. However, my full endorsement of the manuscript will require addressing the following further points.

**Replies to my comments**

- Numerical Analysis. Please specify 'surface curvature, surface growth rate and mesh density were left to default values' in the revised manuscript.

- Exp. details. The added paragraph 'The seven cables from all the cobra probes....to digital converter card' does not address my comment. This section still lacks actual details necessary for repeatability purposes. Please add those in your next review.

- Results. Please include some form of your reply to my question on 'error exist in horizontal shear'.

**Further comments**

- abstract - I suggest replacing 'proper' with 'appropriate'.

- page 2, 65 - I suggest replacing 'cause' with 'can cause'.

- page 2, 65 - I suggest replacing 'All these together' with 'All these phenomena together'.

- page 3, 95 - I suggest replacing 'This standard' with 'The ICE'.

- page 4, 110 - I suggest replacing 'at figure 1' with 'in figure 1'.

- figure 3 - I do not see what this figure adds to the manuscript. I would suggest removing it altogether.

- page 12, 235 - I suggest replacing 'According to Figure 5a &b showing the relative errors between velocities at each height, the largest disconformities....' with 'Figure 5a &b show the relative errors between velocities at each height. The largest disconformities....'

- page 12, 245 - I suggest replacing 'at figure 6' with 'in figure 6'. Please rephrase this throughout the manuscript.

- page 16, 300 - The mismatch in Re is substantial, which is understandable. The authors should limit their discussion on convincing their audience that this is not a strong limiting factors of their work, rather than including general statements on how to increase the Re.

- figure 9 - I remain convinced that figures 1 and 9 should be collated. The blue line can be described at first, then the dashed red line discussion can be left to page 18.

- page 19, 345 - This sentence is unclear. I suggest rephrasing it for clarity.

- page 21, 355 - This sentence is unclear. I suggest rephrasing it for clarity.

- page 24, 395 - I suggest replacing 'As mentioned earlier' with 'As previously discussed'.

- page 24, 395 - The first sentence starting with 'However' is unclear. I suggest rephrasing it for clarity.

- page 28, 405 - I suggest replacing 'with the theory' with 'to the theory'.

- figures 12&13 - The authors insist on not discussing all subfigures, although these are now at least introduced. I have already pointed out that this is far from best practice.

- My concern regarding the lack of a well-structured conclusion section to this work has not been fully addressed. I will not recommend publication of this manuscript without substantial improvements to this section. E.g. page 29, 420 - The first paragraph needs rephrasing for clarity. Page 29, 420 - When the authors say 'despite our simplified assumptions for developing the CFD model', do they mean 'which highlights the limitations of our idealised CFD modelling'? Page 29, 425 - The authors state '...more distorted...'. What does this refer to?

  Please do review this section thoroughly.

---

## Referee Report (RR3)

[referee-annotated manuscript omitted]

---

## Author Response (AR2)

1. **Introduction**

- From my understanding, the EOG and the EWS are not thrown out in the 4th edition but the standard has been expanded. This should be clarified.
- Ans: This is correct. We have addressed this in text (line 35 in marked-up version)
- The wind speed information you are giving in l. 74 is not accurate. If you are bringing up the specific values, you should explain what the values you are just citing here actually mean - not a normal inflow velocity of 50 m/s but the reference wind speed (for the definition, please look into IEC-61400-1 4th edition)
- Ans: clarification has been added (Line 79 in marked-up version)
- While you are now citing the 3rd and 4th edition of the IEC 61400-1 standard, you are using the formulas of the 2nd version (at least, I suppose you do since you did not change the formulas). Change the formulas or the citation. Do not use wrong citations.
- Ans: True! it was a mistake and all the equations and figures have been addressed according to 3rd version. Thank you for this!
- the authors should point our why the gust slicing effect is a problem (recurring high loads while the turbine blade crosses the gust several times)
- Ans: this has been added in line 49 in marked-up version. Thank you!

2. **Methodology**

   **2.1. windEEE dome**

- in the text, both figure 2a and figure 2b are said to show "the closed-circuit 2D ow mode" while the caption says "render of test chamber and ow path" (a) and "side view schematic of the WindEEE Dome with ow path in closed-circuit straight ow mode" (b). Please clarify the description. Do I interpret the figures correctly when saying that the flow returns around the inner chamber (i.e. left, right, above)?
- Ans: this has been addressed now as figure 1. The flow just recirculates from top, the surrounding part of the outer shell are isolated from the fans' suction side with special sealed doors in 2D flow mode. The figure has been reconfigured accordingly.
- you are mentioning the repeatability of gusts and shear flows. I would like to know whether this has been shown somewhere and if a citation could be included.
- Ans: we have not done any repeatability study on the produced gusts yet, as now specified in manuscript (line 219 in marked-up version); we tried various actuating time for fans and IGVs then processed the data and selected the best configurations for which the results are presented. In order to recognize this we deleted the word "repeatable" in line 148 even though the gusts from IGVs look consistently identical (figure 11f & g).

   **2.2. Numerical Flow Analysis Setup and Tuning/Validation**

- In this part, you should be clear about using one static condition to validate the numerical setup while the numerical setup will afterwards be used dynamically to generate the gusts and the shear. This could also be used in the discussion of the deviation in the flow field due to the

different performance in the upper fan rows (as you write, "By knowing these discrepancies, corrections can be applied to the fan inputs").

- Ans: This has been added in the manuscript, thank you! Line 204, 310.

**2.3. Gust length and time scaling**

- I could not find any reference of the 14s duration of the 50 year EOG in the IEC 61400-1 ed 3 and 4 standards, could you please give me the section where I can find this?
- Ans: That was based on second version, it has been addressed in line 228 and other relative sections and figures.
- (question out of interest): with a TSR of 1.1 and an inflow velocity of 5 m/s, the rotational frequency of the turbine (R = 1.1m) would be less than 1Hz which appears very slow to me. Is there a mistake?
- Ans: True: the RPM will be around 56. To capture the effect of the 5 s gust on the turbine and being able to relate it to the full scale based on our assumption that should be the operational TSR for a future experiment that would include the turbine.
- It is not clear what you want to express here. Please rephrase this.
- Ans: it has been rephrased in Line 263. This part presents the assumptions for the parameters that affect the profile of the extreme events. the full scale and scaled extreme condition figures have been moved together in this section.

**3. Results and discussion**

**3.1. Steady wind shear**

- I would change the wording in line 263: you are talking about velocity deviations, but the standard deviation gives you a measure of the fluctuations in the flow

- Ans: it has been addressed in line 298.

- it is not clear what the setting of 1.6 s is and what it implies. If the information is relevant, please be more specific.

- Ans: explanation has been added in line 323.

- I do not agree with the use of the term "intermittency" in this context because in turbulence, it is normally used for effects occurring in the ow due to the turbulence while here, in contrast, the short, strong dip in the velocity is not caused by turbulence but more likely due to the setup. I suggest rephrasing the explanation.

- Ans: Different wording has been used in lines 362 and 364.

- A thought regarding the velocity "dip": When looking at the time series, it is obvious that the higher measurement positions have higher turbulence levels and in addition, probe H measures this "dip" - could this be explained by the upper fans having a less stable air supply than the lower fans who will suck more air from the sides and above? (due to the sharp angle between

the upper row of the fans and the plenum, I would expect more air to ow from there to the lover fans) And could this dip be related (lack of air when suddenly increasing the velocity)?

- Ans: That is overall correct. The upper fans have less supply due to the sharp angle. However, this effect has been minimized in WindEEE by installing returning vanes at the top of the outer circuit. We have now commented on this aspect in line 363 in marked up version.

**4. Conclusion**

- While you are talking about a "numerical and experimental study" both in your title and the discussion, I do not feel that this is an accurate description since you are describing how you build a numerical "setup-preparation chain" that you validate against previous ABL data sets, and this numerical setup is in the following used to set the vans. However, you are not showing the simulation results of the gusts and shears, and therefore it is not directly a numerical study in my opinion. What you compare are the experimental results and the expected results from the formulas. I therefore think it would aid if you would rephrase the first sentence of the conclusion. Is a numerical study with a simulation and experimental campaign for example with the turbine planned in the future?
- Ans: line 381 has been modified to recognize this. The intent going forward is indeed to be able to compare simulation to experiment.
- "The steady experiment runs corresponding to the peak of the shear cases show the fans act non-linearly and they have different individual efficiencies, especially the top and bottom rows despite our simplified assumptions for developing the CFD model." I would suggest rephrasing: "The steady experiment runs corresponding to the peak of the shear cases show THAT the fans act non-linearly and THAT they have different individual efficiencies, especially the top and bottom rows. This has not been taken into account in our simplified assumptions for developing the CFD model."
- Ans: thank you, line 415 has been modified.

**5. Referee comments**

- It would have been nice if specific information (for example the lines) on the changes that have been made in response to the reviewers' comments would have been added.
- Ans: Thank you for noting that, we have tried to highlight the corresponding line to the modifications made in the marked-up version along with the line numbers that are included here.
- When asking about a) the phase average of your gust and b) the repeatability, comment on fig. 12, I wanted to know whether you repeated the experiment several times and checked a) whether the gust events are similar and b) whether the plots are presenting an average over multiple gusts. I did understand that you use a filtering with a 0.2s moving average but this is not a phase average. Could you please clarify the above-mentioned points and add this in the paper? I could not find the information that the cobra probes show variations in the measurement of the mean velocity in the revised paper.
- Ans: The plots have not been phase averaged based on multiple experiment runs. They are just showing one individual experiment run as now specified in manuscript (line 219 in marked-up

version). We tried various actuating time for fans and IGVs then processed the data and selected the best configurations for which the results are presented. In order to recognize this we deleted the word "repeatable" in line 148 even though the gusts from IGVs look consistently identical (figure 11f & g). A study on the reproducibility has been recommended for future work (line 415). The velocity variations due to the filtering has been added in line 339.

**RC2 Comments,**

1. **Referee comments**

- Numerical Analysis. Please specify `surface curvature, surface growth rate and mesh density were left to default values' in the revised manuscript.
- Ans: they have been added in line 169 in marked-up version
- Exp. details. The added paragraph `The seven cables from all the cobra probes....to digital converter card' does not address my comment. This section still lacks actual details necessary for repeatability purposes. Please add those in your next review.
- Ans: as has been added in line 219, no repeatability investigation has been done yet. Because we did not know how to modulate the actuating time of the power and IGVs, a set of experiments with different timing were run. We selected the best configurations for which the results are presented. In order to recognize this we deleted the word "repeatable" in line 148 eventhough the gusts from IGVs look consistently identical (figure 11f & g).
- Results. Please include some form of your reply to my question on `error exist in horizontal shear'.
- Ans: it has been added in line 310.

2. **Further comments**

- abstract - I suggest replacing `proper' with `appropriate'.
- Ans: addressed at line 18.
- page 2, 65 - I suggest replacing `cause' with `can cause'.
- Ans: addressed at line 48
- page 2, 65 - I suggest replacing `All these together' with `All these phenomena together'.
- Ans: addressed at line 67.
- page 3, 95 - I suggest replacing `This standard' with `The ICE'.
- Ans: addressed at line 73
- page 4, 111 - I suggest replacing `at figure 1' with `in figure 1'.
- Ans: it has been addressed through the manuscript
- figure 3 - I do not see what this figure adds to the manuscript. I would suggest removing it altogether.
  Ans: Thank you for your comment but the gusts with IGVs had the best results; we feel a small presentation of them helps the audience to understand their function.
- page 12, 235 - I suggest replacing `According to Figure 5a &b showing the relative errors between velocities at each height, the largest disconformities....' with `Figure 5a &b show the relative errors between velocities at each height. The largest disconformities....'
- Ans: thank you! It has been changed in line 195.

- page 12, 245 - I suggest replacing `at figure 6' with `in figure 6'. Please rephrase this throughout the manuscript.
- Ans: this has been done, thanks.
- page 16, 300 - The mismatch in Re is substantial, which is understandable. The authors should limit their discussion on convincing their audience that this is not a strong limiting factors of their work, rather than including general statements on how to increase the Re.
- Ans:  It has been moved to line 282.
- figure 9 - I remain convinced that figures 1 and 9 should be collated. The blue line can be described at first, then the dashed red line discussion can be left to page 18.
- Ans: They have been moved together as figure 8 in section 2.4. thank you!
- page 19, 345 - This sentence is unclear. I suggest rephrasing it for clarity.
- Ans: it has been addressed in line 307.
- page 21, 355 - This sentence is unclear. I suggest rephrasing it for clarity.
- Ans: it has been addressed in line 321.
- page 24, 395 - I suggest replacing `As mentioned earlier' with `As previously discussed'.
- Ans: it has been addressed in line 358.
- page 24, 395 - The first sentence starting with `However' is unclear. I suggest rephrasing it for clarity.
- Ans: it has been addressed in line 362.
- page 28, 405 - I suggest replacing `with the theory' with `to the theory'.
- Ans: it has been addressed in line 379.
- figures 12&13 - The authors insist on not discussing all subfigures, although these are now at least introduced. I have already pointed out that this is far from best practice.
- Ans: figure 12 now as 11, it is just showing the total time history related to the whole experiment, we have tried to add more discussion about this figure. Figure 13 now as 12, which is more detailed; we have added to our discussion in lines 360 to 376 to this section. Thank you for your constructive comments.
- My concern regarding the lack of a well-structured conclusion section to this work has not been fully addressed. I will not recommend publication of this manuscript without substantial improvements to this section. E.g. page 29, 420 - The first paragraph needs rephrasing for clarity.
- Ans: the conclusion has been restructured and improved.
- Page 29, 420 - When the authors say `despite our simplified assumptions for developing the CFD model', do they mean `which highlights the limitations of our idealised CFD modelling'?
- Ans: Yes! That sentence has been rephrased.
- Page 29, 425 - The authors state `...more distorted...'. What does this refer to?
- Ans: It was rephrased in Line 418. By distortion I meant the time lag between the high and the low peak hitting the test section.
- Please do review the conclusion.
- Ans: We restructured the conclusion.

[revised manuscript text omitted]

---

## Author Response (AR3)

- The section on experimental details lacks some of the details that one would like to see in a high-quality paper (i.e. details on setup on the probes, justification of sampling frequency and time, a/d conversion details, the measurement stations, and many other). I have pointed this out before, however, my recommendations were largely ignored/misunderstood. I suggested these details were needed to allow other research groups to replicate the experiments. I would still like to see some of these things added to the paper for this purpose.

Ans: As now has been added, the precise turbulence characteristics are not the primary goal of this study so the sampling frequency was roughly chosen based on previous studies performed in the facility. [Refan, M. and Hangan, H.: Near surface experimental exploration of tornado vortices, J. Wind Eng. Ind. Aerodyn., 175, doi:10.1016/j.jweia.2018.01.042, 2018] [Romanic, D., LoTufo, J. and Hangan, H.: Transient behavior in impinging jets in crossflow with application to downburst flows, J. Wind Eng. Ind. Aerodyn., 184, doi:10.1016/j.jweia.2018.11.020, 2019.]

The sampling duration was chosen long enough to make sure the flow recovers, and we will not see any effect of the extreme event in the recirculated flow, (one complete recirculation takes about 12 seconds based on Windeee size and the average wind velocity), these have been added in line 202 in marked up version. The a/d conversion and cobra probe connections are straight forward, and all came in one package that was bought from the Turbulent Flow Instrumentation company. Detail about the cobra probes connection with a new subfigure (figure6d) has been added to the manuscript. (Line 205 to 2016 in marked up version). Hopefully you find these new details satisfactory.

**RC2 Comments,**

Overall, the quality of this manuscript has significantly improved after two major revisions. Nevertheless, there are some improvements that should be made (see also comments in the attached PDF (comments are visible using the adobe acrobat reader or okular).

- the document should be checked again for grammatical errors and typos - some things are marked in the attached PDF

Ans: The comments in the pdf file has been addressed. lines 41, 51, 60, 72, 77, 80, 84, 96, 172, 267, 298, 313, 332, 387, 425, 430 in marked up version.

- ll.28 probably, you could rephrase that paragraph since currently, the question arises why you are using the IEC gust when it is far away from a real gust

Ans: That sentence is preparing the audience for the reason to the new version of the standard. "It has also motivated the most recent edition of the IEC standard" then the following sentence" However, the third edition of the IEC standards was used in the work presented here as an initial step towards gust experimentation and represents an incremental development of a gust loading experimental capability".

- ll.208 since your TSR is very low, you are already in the order of magnitude of 4 rotor revolutions during the 5s gust and the question arises to me why it is necessary to introduce the gust propagation time. It would help to motivate why this is more appropriate. Also, the sentence "For a scaled wind

turbine in the wind tunnel 4 rotor revolutions happen on the order of a second at the nominal operating condition." is not true for your setup, so you should rephrase it.

Ans: that sentence mentions **nominal** operating condition in wind tunnels, 4 revolutions of rotor happen in a second. Then we relate that to the propagation time of the tip vortex so we can adjust the relevant time duration for gust based on other parameters like TSR and wind velocity. In our experiment the turbine will not work in its nominal condition.

- ll. 289 please clarify whether you switched the fan power gradually, "switching fans to 30% for 1.6 s" reads like there was no gradual increase.

Ans: we input sudden changes in the software. Different set of wordings were used. Line 307 in marked up version.

- for your whole argumentation, it would be favorable to plot/add the simplified gust in fig. 12 d-g since you are aiming to match the simplified gust and not the IEC gust with your control - the results will match much better.

Ans: in all the EOG figures, simplified gust has been replaced. Corresponding changes in the text has been performed.

For future publications, please make sure your manuscript has a good quality, structure and use of language when handing it in for the first time.
Also, when publishing more on the capability of the WindEEE Dome about the extreme wind conditions, I would recommend
- verifying the reproducibility of events
- using a phase average (+filter if necessary) to have a smooth gust information
- showing the unprocessed data so that the fluctuations that are currently averaged out are also presented to the reader
- commenting on all velocity components: you do have the information and if you can show that there are no/small v and w components, this is also an important information.
- splitting the figures. It is really hard to look at a figure that extends over several pages. It is much easier to have a figure for the shear, one for the EOG with fan power and one for the EOG with blockage where you can directly take the information from the caption.

Ans: we highly appreciate your constructive comments. These will be considered in our future works. Regarding splitting the figures in Figure 11 and 12, the caption for all of them is similar and it just mentioning the blue line is from experiment and the orange line is from the standard.  Splitting them would have made a lot of redundancy. But we will try to come up with a better way of presentation in future.

[revised manuscript text omitted]